# A model based on Bayesian confirmation and machine learning algorithms to aid archaeological interpretation by integrating incompatible data

Daniella Vos[1¤]*, Richard Stafford[2], Emma L. Jenkins[1], Andrew Garrard[3]

**1** Department of Archaeology and Anthropology, Faculty of Science and Technology, Bournemouth University, Poole, United Kingdom, **2** Department of Life and Environmental Sciences, Faculty of Science and Technology, Bournemouth University, Poole, United Kingdom, **3** Institute of Archaeology, University College London, London, United Kingdom

¤ Current Address: Department of Cultural Geography, Faculty of Spatial Sciences, University of Groningen, Groningen, The Netherlands

* d.vos@rug.nl

## Abstract

The interpretation of archaeological features often requires a combined methodological approach in order to make the most of the material record, particularly from sites where this may be limited. In practice, this requires the consultation of different sources of information in order to cross validate findings and combat issues of ambiguity and equifinality. However, the application of a multiproxy approach often generates incompatible data, and might therefore still provide ambiguous results. This paper explores the potential of a simple digital framework to increase the explanatory power of multiproxy data by enabling the incorporation of incompatible, ambiguous datasets in a single model. In order to achieve this, Bayesian confirmation was used in combination with decision trees. The results of phytolith and geochemical analyses carried out on soil samples from ephemeral sites in Jordan are used here as a case study. The combination of the two datasets as part of a single model enabled us to refine the initial interpretation of the use of space at the archaeological sites by providing an alternative identification for certain activity areas. The potential applications of this model are much broader, as it can also help researchers in other domains reach an integrated interpretation of analysis results by combining different datasets.

## Introduction

The archaeological record comprises various forms of artefacts and ecofacts which, when studied in combination, offer a better interpretative understanding of past human lifeways, than when studied in isolation. The range of archaeological material that can be studied, and the amount of information that can be gained from it, is dependent upon the original concentration of material deposited into the archaeological record by its human inhabitants and the state of preservation of the archaeological site itself. Ephemeral sites and/or their poor

**Funding:** This research was supported by an AHRC/Bournemouth University (BU - https://www.bournemouth.ac.uk/) PhD studentship and by an Arts and Humanities Research Council (AHRC - https://ahrc.ukri.org/) grant number AH/K002902/1 awarded to ELJ. The funders had no role in study design, data collection and analysis, decision to publish, or preparation of the manuscript.

**Competing interests:** The authors have declared that no competing interests exist.

preservation represent a challenge that archaeologists can embrace by finding novel and innovative ways to maximise the amount of information that be gained from the archaeological record.

In recent years, the interpretation and understanding of the use of space in archaeology has been increasingly aided by the incorporation of geoarchaeological techniques, such as geochemistry, phytolith analysis, micromorphology and lipid residue analysis alongside artefactual analysis [1–8]. These techniques have the advantage of considering *in situ* signals of activity, which are less prone to post depositional disturbance [4, 9]. The downside, however, is that they may produce results which are equivocal, subtle, or distinct, which can limit their interpretation potential. Therefore, while these methods greatly increase our understanding of archaeological sites, their interpretative value is greater when used in combination as a multi-proxy approach than in isolation [2].

Inconsistencies in the type of results produced by different techniques and their resulting data structure, however, often do not favour their incorporation within a single comprehensive statistical model [10, 11]. For example, while phytolith data is recorded in count form and may be further aggregated into several categories representing plant genus, plant parts or weight of phytolith material per gram, measurements of geochemical elements are often recorded in parts per million, producing continuous data. Many geoarchaeological techniques produce complex, high dimensional data which are difficult to interpret in relation to one another. The results of multiple analysis techniques are therefore often quantitatively analysed separately even when they are used within a combined methodology, and then described side by side to form a qualitative synthesis [7, 11–15]. One way to approach this issue is through standardization and normalization of the data prior to their integration in multivariate statistics [16–18]. However, while the scale of the data is brought to the same, continuous level, very different phenomena are measured which do not necessarily all translate well into a continuous scale.

This paper offers a new approach to integrate analysis results from multiple sources of information through the application of machine learning algorithms alongside other statistical and traditional analytical methods within a single quantitative framework. It is different to previous ones in that a hypothesis, or the probability of an interpretation, can be quantitatively addressed by the incorporation of almost any type of information. Its use is illustrated through a case study where a combination of geochemistry and phytolith analysis was used to aid the identification of activity areas in ephemeral archaeological sites.

The two datasets were combined by means of Bayesian confirmation to examine whether such a combination is likely to enhance the predictive power of the techniques compared to their use in isolation. Bayesian confirmation is derived from Bayes' theorem, which is used to deduce properties about a population, or model a probability estimate for unobserved events, based on *a priori* information [19–21]. By extension of this, Bayesian confirmation combines prior information about a population with new evidence gathered through observation or experiment. The evidence is used to either support or contradict a hypothesis. In effect, the confirmation of a hypothesis occurs when the posterior probability (the updated probability of the hypothesis following the consideration of new evidence) exceeds the prior probability (the probability of the hypothesis before the incorporation of new evidence), and the greater the difference–the greater the confirmation [20].

In our case, Bayesian confirmation was particularly useful since it enables combining information from different types of data once we "translate" the results into a form that is meaningful within a probability model. In order to do so, decision trees are used to classify the data. Classification or decision tree algorithms assign variables to discrete classes through binary recursive partitioning. This process splits data into subgroups based on values of predictor variables in order to predict how data will behave based on the current observations [22].

Decision trees are very suitable for "translating" the results of various analysis techniques into conditional probabilities since they divide the data into pre-defined meaningful categories, providing both a way to examine the nature of each sample and an indication of the overall likelihood of a correct classification. This numerical output, in itself a probability, is used as an input for the amount of probability that is either added to or deduced from the prior probability in the model based on Bayesian confirmation.

By applying a framework combining Bayesian statistics and decision tree algorithms, we aim to maximise the information that can be gained from archaeological data and combine the continuous and discrete datasets produced by the two analysis techniques in an effective way. Bayesian confirmation is suitable for this task since it allows one to deal with uncertainty by means of allocating a probability to a particular outcome. Seeing as it is not possible to observe past human behaviour as it takes place, or rely on repetition of experiments, it is not possible to truly test the accuracy of archaeological interpretation. What is possible, however, is to provide it with numerical probability based on multiple sources of evidence. The more evidence pointing towards a particular scenario, the more weight it receives. In this way, we can add a quantitative likelihood to our qualitative interpretation of archaeological features.

## Case study

### Geochemical and phytolith data from Wadi el-Jilat sites

Fieldwork at Wadi el-Jilat (Fig 1) was part of a series of excavations in the Azraq Basin during the 1980s. The project aimed to provide new insights into settlement and subsistence in the steppe and desert regions of the Levant during the period when sedentism, agriculture and pastoralism were first developing in the more fertile regions of the Levant and more generally around the "Fertile Crescent" of South-West Asia [23]. The vast majority of Neolithic buildings at Wadi el-Jilat are circular or oval semi-subterranean constructions, with upright slabs forming the fragile external walls, which often enclosed shallow deposits. Many of these structures had internal divisions, hearths and other features such as benches or storage bins [23, p.40-1]. Nevertheless, unlike contemporary sites in moister regions of the Levant, which present substantial architectural remains, the Neolithic settlements at Wadi el-Jilat left traces of somewhat 'flimsy' structures. These, in combination with the faunal evidence, hint towards a seasonal occupation, as is the case with many ephemeral structures used today by modern nomadic populations [24–28].

The ephemeral nature and long abandonment of these sites hindered the interpretation of the past use of space by their inhabitants, requiring additional means to confirm the initial identification of activity areas in the field. To that end, eighteen samples from the Pre-Pottery Neolithic B (PPNB) site of Wadi el-Jilat 7 (WJ7) and thirty-one soil samples from the Early Late Neolithic (ELN) site of Wadi el-Jilat 13 (WJ13) were analysed for their phytolith and geochemical soil signatures as part of a study into the potential of a dual geochemical-phytolith methodology to aid the interpretation of ephemeral archaeological sites [15, 29]. The soil samples were chosen to represent a range of occupation deposits with the aim of comparing between soil signatures across various context categories (described in Table 1). WJ13 provided a suitable case study for the application of Bayesian inference to aid the interpretation of archaeological data, because unlike the neighbouring site of WJ7, the results of the geochemical and phytolith analyses did not entirely comply with the division into the context categories defined in the field.

**Geochemistry data.** Within an archaeological context, soil chemistry is used to link patterns of enrichment or depletion of certain chemical elements in the soil to past human occupation [30, p.36]. The application of geochemistry to archaeological sites is often used to locate

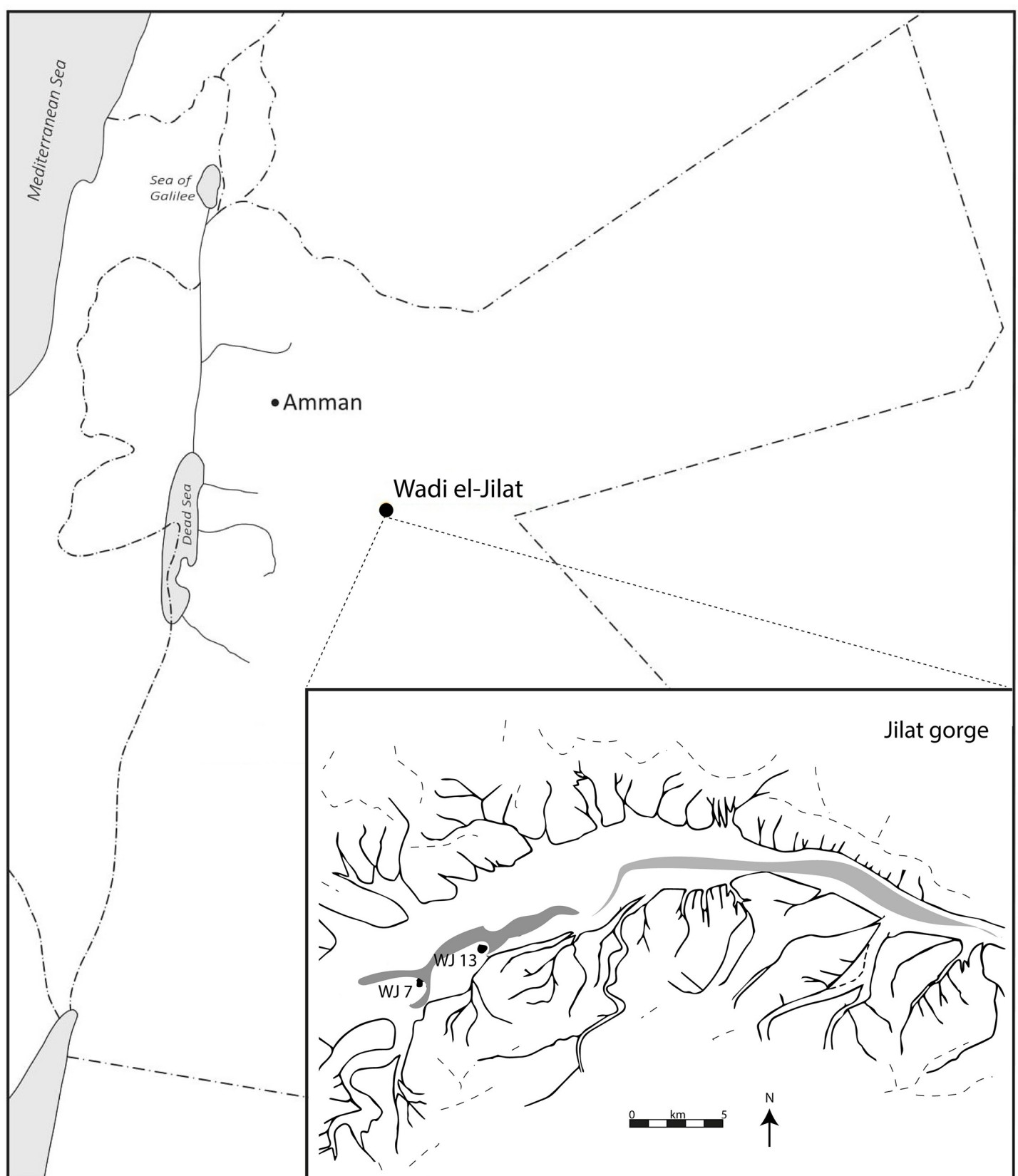

**Fig 1. Location of the Wadi el-Jilat sites.** A close up of the Wadi el-Jilat gorge includes an overview of the individual site locations.

**Table 1. Categories of soil samples from Wadi el-Jilat sites used for the geochemical and phytolith analysis in this study.**

| Categories used in this study: | Explanation: |
| --- | --- |
| Deposit | General deposit, fills within and between buildings |
| Activity area | Locus rich in bone and lithic material |
| Hearth | Deposit consisting of ashy sediment |
| Compact ashy fill | Compact fill containing traces of ash |
| Bedrock feature | Posthole or other man made pits in bedrock |
| Other | Features that do not fit the previous categories, this category mainly consists of pits and bins |
| Background | Background sample taken from the vicinity of the site |

and delineate settlements, refuse areas, graves, agricultural plots and production areas. It can also be applied on a site level to obtain a better understanding of stratigraphy and sedimentology, or help interpret the distribution of activity areas and features [29, 30]. The geochemical analysis at the Wadi el-Jilat sites (WJ) was performed using a Thermo Scientific Niton XL 3t Goldd+ (geometrically optimised large area drift detector) handheld XRF (X-ray fluorescence) analyser. The geochemical interpretation in this study focused on the following chemical elements, measured in parts per million (PPM): aluminium (Al), calcium (Ca), iron (Fe), potassium (K), magnesium (Mg), chlorine (Cl), manganese (Mn), phosphorus (P), strontium (Sr), titanium (Ti), sulphur (S), zinc (Zn), chromium (Cr), and zirconium (Zr) (see 48 for an overview of the geochemical analysis protocol).

The site of WJ7 did not appear to show any remarkable trends when comparing the enrichment of individual chemical elements across context categories (for detailed overview of geochemical results see 47 or 49). The site of WJ13 did portray more variation in enrichment of chemical elements between context categories, but to a limited degree. The most prominent distinction between activities appeared to be represented at this site through P, levels of which were increased in all anthropogenic contexts in comparison to the background samples, noticeably mostly in the posthole samples (Fig 2). This could be explained by leaching of P downwards, but then one would expect to see a similar pattern in the other Wadi el-Jilat sites, which is not the case [29]. There is a slight elevation of K and Mg in the hearths, and of Mn in activity areas (Fig 2). These trends are similar to observations made at ethnographic sites in a related study [11].

Contrary to the visibility of enrichment of individual elements across context categories, the 3D biplot created for the principal component analysis (PCA) for WJ7 provided a better result than the one produced for WJ13, explaining 89% and 72% of variance respectively, and presenting better clustering into the excavator's context categories (Figs 3 and 4). The geochemical variables that drive most of the variance within the PCA for the site of WJ7 (Fig 3) seem to correspond, at least partly, with the elements that were found to indicate anthropogenic input in earlier studies, mainly Mg, Fe, S, P, K, Ca and Cl (Table 2 provides an overview of chemical elements found at the sites of Wadi el-Jilat and their suggested association with anthropogenic anomalies in earlier studies). The geochemical elements driving the variance for the PCA biplots created for the site of WJ13, however, do not correspond as well with findings from previous studies (Fig 4). While still relying on a few anthropogenic indicators which were found to be relevant in other studies (Fe, K, Mg, Ca, P, and Cl), much of the variance is driven by chemical elements which are not considered to be significant anthropogenic indicators (Ti, Si, Al, Zr, Nb, Rb and V). WJ7 and WJ13 portray different patterns of enrichment of geochemical and phytolith soil signatures which might suggest a difference in their occupation

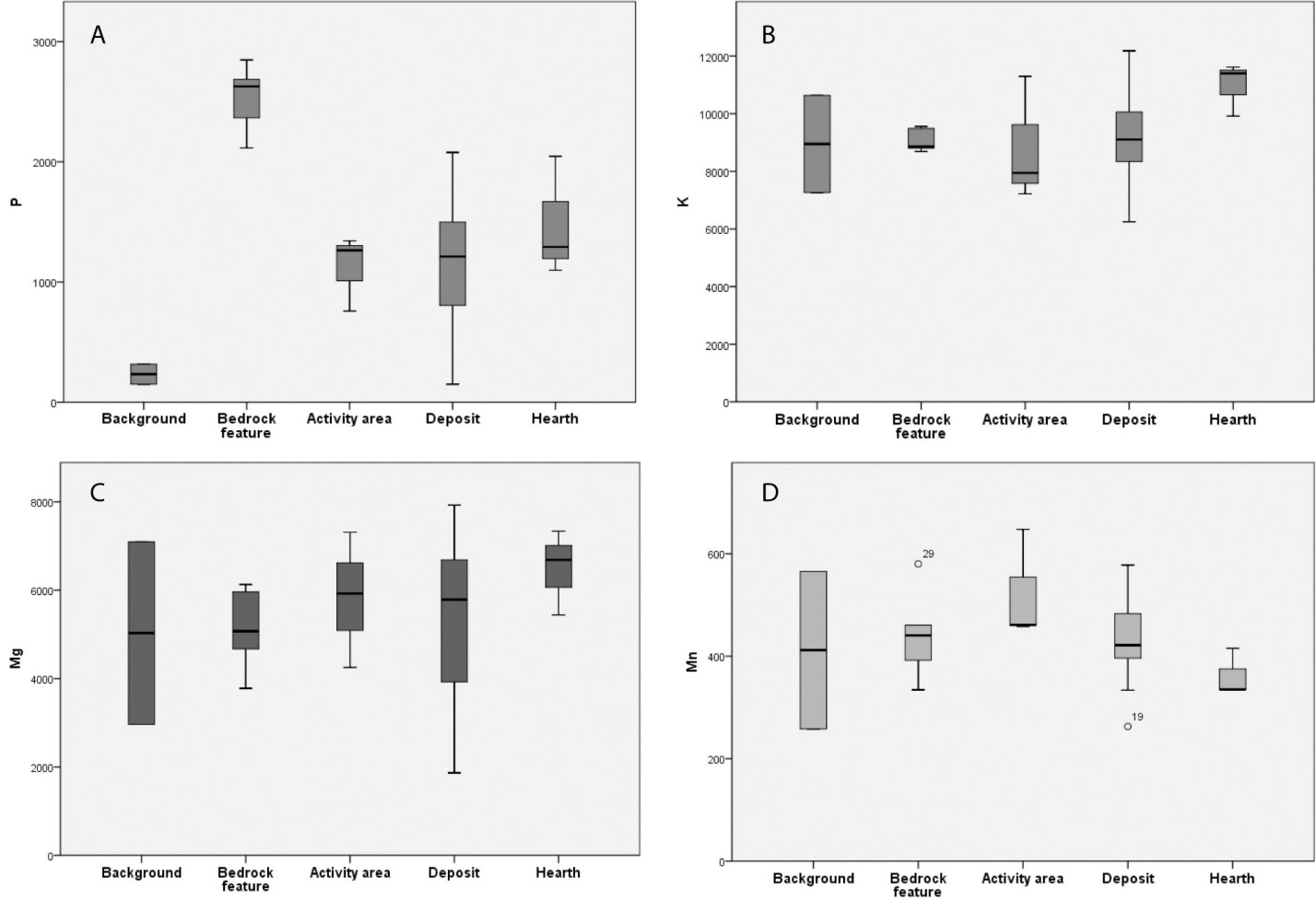

**Fig 2.** Average measurements in ppm for all samples within WJ13 per context category for the following chemical elements: (A) P, (B) K, (C) Mg and (D) Mn.

sequences, their use, the influence of local taphonomic processes, or alternatively the degree of successful identification of their features in the field (which influences the division of soil samples into suitable context categories).

**Phytolith data.** The word phytolith in archaeological context generally refers to opal or amorphous silica representations of plant cell structures, of either single cells or conjoined cells [6, 37]. Single-celled phytoliths can be classified according to their morphologies, which often link to different plant parts (for example husks or leaves) and can thereby indicate patterns of plant use (such as plant-processing). In addition, both multi- and single-celled phytoliths can distinguish between two groups of plants: monocotyledons (monocots) and dicotyledons (dicots–here we use the term according to the pre 1990s definition to mean non-monocotyledon) [38, 39]. By adding up the information from a variety of phytolith forms and morphologies within a sample, a profile of plant use at a site can be created.

The extraction of phytoliths from the WJ soil samples was carried out using the dry ashing method [40]. The names of the phytolith types followed the International Code for Phytolith Nomenclature [41]. In addition to phytoliths, diatoms and sponge spicules were recorded. The phytolith database included the morphological categories used in the counting sheets and additional variables calculated from the raw data: dicot, monocot, single-cell, multi-cell,

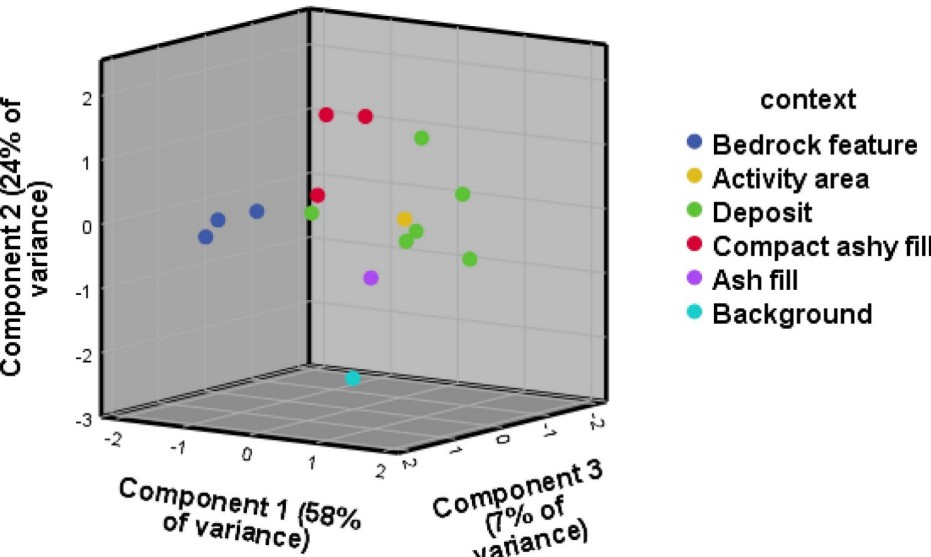

**Fig 3. PCA biplot for the geochemical results, WJ7.** The first component is driven by Mg, Si, Ti, Fe, S, Zr, K and P, and the second component by Ca, Sr and Rb. The third component is driven by Cl.

Panicoideae, Pooideae, Chloridoideae, Arundinoideae, Palmaceae, *Hordeum sp.*, *Triticum sp.*, leaf, leaf/husk, leaf/stem, husk, awn, weight percent of extracted phytoliths, and number of phytoliths per gram of original sediment processed (for a complete overview of the phytolith analysis and recording methodology see 47).

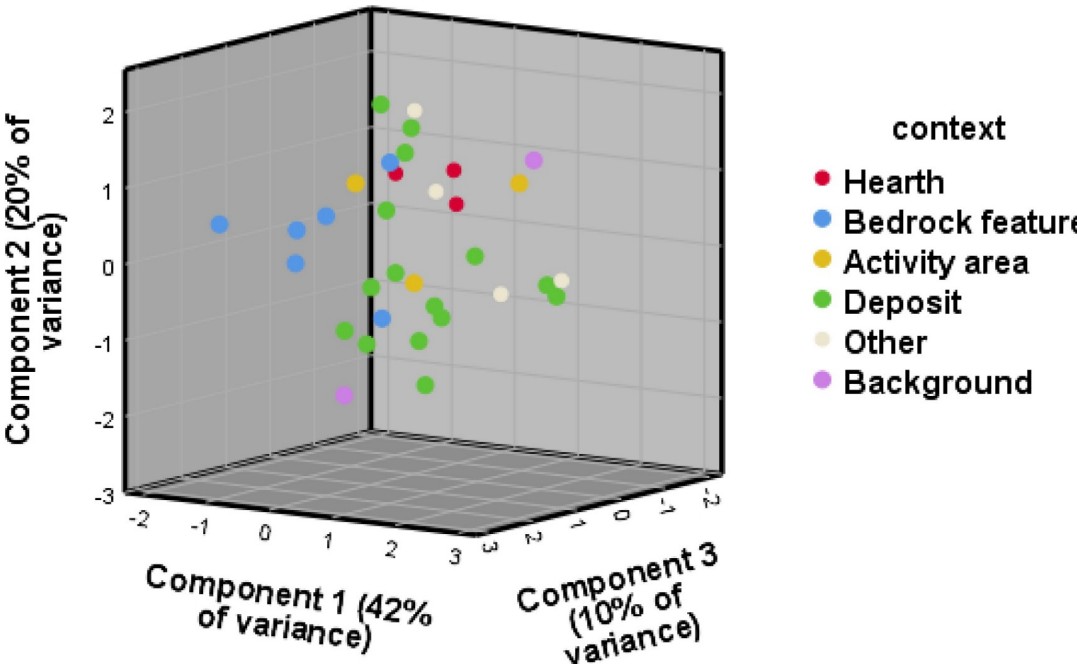

**Fig 4. PCA biplot for the geochemical results, WJ13.** The first component is driven by Ti, Si, Fe, K, Al, Zr and Nb. The second component is driven by Mg, Ba, Sr and Ca, and the third by Cr, P, Rb, Cl and negatively by V.

Table 2. Associations between chemical elements and anthropogenic related activities found in earlier studies and, where applicable, anomalies found at the Wadi el-Jilat sites [after 29, p.271-7].

| Chemical element | Associated activity in previous studies | Associated activity in this study |
|---|---|---|
| P | Hearths, animal dung [11]; food preparation and consumption [31–33], burning and food storage [34], refuse areas [31], excrements [33], Byres [35], Meat [36] | General anthropogenic occupation (all WJ sites), bedrock features (WJ13) |
| Mg | Hearths, animal dung [11]; wood ash [5], cooking hearths, food preparation and consumption [31], Meat [36] | Hearths (WJ13) |
| Ca | Hearths [11, 31], food storage and preparation [33, 34], lime use? [5] | General anthropogenic occupation (all WJ sites) |
| K | Hearths, animal dung [11]; wood ash [5], cooking hearths, food preparation and consumption [31] | Hearths (WJ13) |
| Mn | Hearths/burning [11, 34], vegetable [36] | Activity areas (WJ13) |
| S | Hearths, animal dung [11] | Hearths and bedrock features (WJ7) |
| Sr | Hearths [11, 35], excrements and food preparation [33], Lime use? [5] | Slight elevations in hearths (all WJ sites) |
| Cl | Animal dung, hearths, animal pens [11] | |
| Fe | Background [11]; Craft production (high levels in combination with burning) [34], burning [33] | |
| Ti | Background [11, 34] | |
| Al | Background [11, 34] | |
| Cr | Not measured in previous studies | Bedrock features (WJ7, WJ26) |

The results of the phytolith analysis at Wadi el-Jilat revealed only very subtle patterns of differentiation between activity areas within the sites, while the background samples were clearly different to the on-site material. A high monocot to dicot ratio, an abundance of grass husks and a high weight percent and number of phytoliths per gram all appear to be associated with anthropogenic activity at the Neolithic sites [29]. The phytolith analysis results at WJ7 demonstrate the most variability in context categories. While all contexts show an increase of monocots in relation to the background samples, the categories 'activity area' and 'compact ashy fill' (which probably reflect hearths) contain the highest concentrations of these. The two context categories show resemblance when it comes to plant parts, containing the largest amounts of husk material in relation to the other context categories (Fig 5).

The enrichment of silica aggregate material at the bedrock features, hearths and deposits of WJ13 might also indicate a high anthropogenic input, albeit of a different kind (Fig 5). Silica aggregate is considered to be derived from woody material, and could indicate the presence of wood or charcoal within these features [42, 43]. Nevertheless, when comparing the results among the different context categories, the individual phyolith trends at WJ13 do not portray the expected anthropogenic enrichment input identified for WJ7 and other similar sites [7, 8, 29, 44–46]. The PCA 3D biplots created for the phytolith results at the WJ sites do portray some clustering, but to a limited degree (Figs 6 and 7). As with the results of the geochemical analysis, the second and third components represent less of the overall variance but demonstrate better clustering of context categories than the first two components.

## Methods and results

### Decision trees

Decision trees were used in this study to understand and visualise how well the quantitative data from phytolith and geochemical samples are categorised into the pre-defined activity

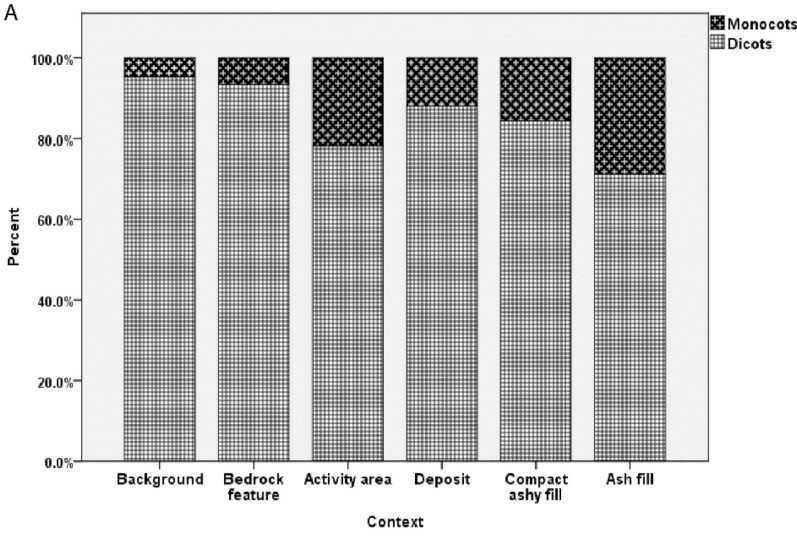

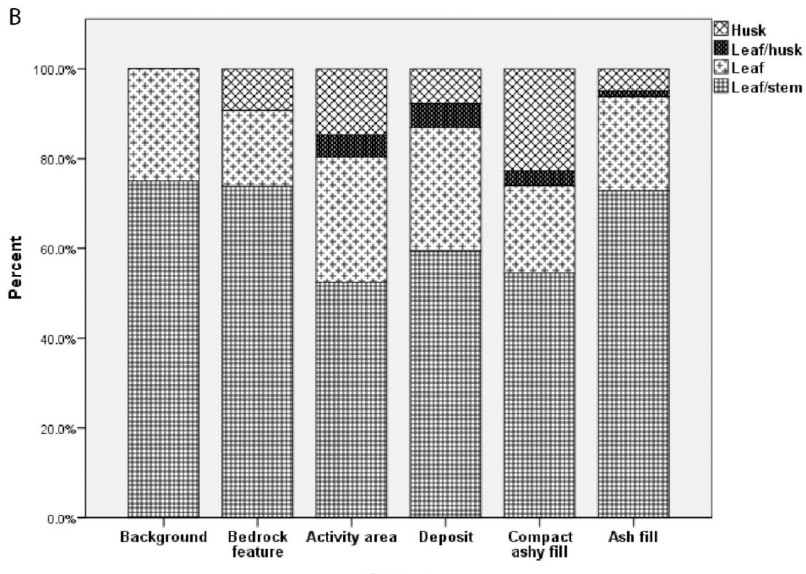

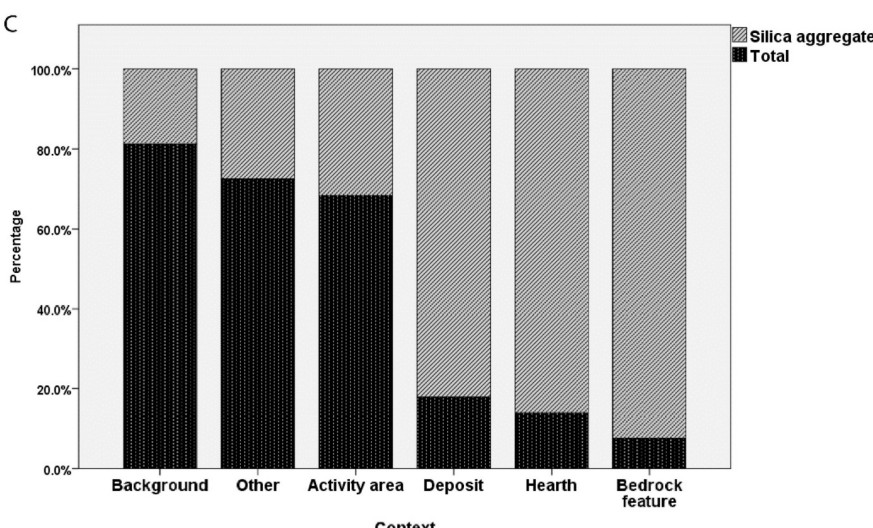

**Fig 5.** A) Ratio of monocot to dicot per context category at WJ7; B) Plant part ratio per context category at WJ7; C) Silica aggregate and phytolith count ratio per context category at WJ13.

areas, and which variables are important within this classification. Decision tree algorithms assign variables to discrete classes through binary recursive partitioning, a process which splits data into subgroups based on values of predictor variables.

The decision trees were created in Weka 3.6.13, a software package with a range of classification and prediction machine learning algorithms. The J48 classifier was used in this analysis, a decision tree classification algorithm based on a top-down recursive divide-and-conquer mechanism [22, 47, 48]. An attribute is selected for the top (root) node, and a branch is created for each possible attribute value. The process splits the instances into subsets (one for each branch). The process is repeated recursively for each branch until all instances have the same class. The desired outcome is the smallest possible decision tree with pure nodes.

There are many different possibilities for carrying out a similar classification which would result in similar outcomes–a classification into pre-defined categories with the associated probabilities of a successful classification which are needed as input for the Bayesian calculation (which is described in the following section). However, the J48 classifier produces a well pruned tree, meaning that a particular probability of classification will be conservatively estimated [49]. This conservative estimate is useful when used as evidence in Bayesian confirmation to prevent over inflation of final, posterior, probability estimates.

In our case, each soil sample was identified in the field as a particular activity type, based on the excavators' interpretation of the relevant locus/context (Table 1). The data for each sample, including a context category and the results of the soil analyses (see description of datasets above), provided the input for the decision tree algorithm. The values for the chemical

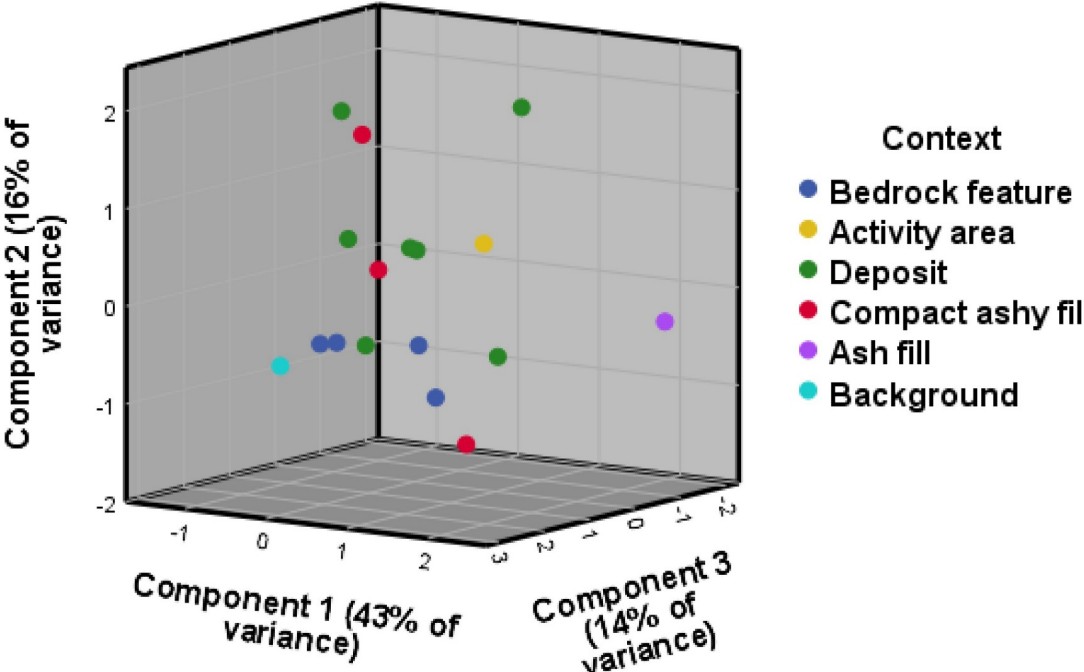

**Fig 6. PCA biplot for the phytolith results, WJ7.** The first component is driven by monocots, unidentified and degraded phytoliths, leaf, leaf/stem, Pooideae and single-cell phytoliths. The second component is driven by weight percent, Chloridoideae and negatively by burnt phytoliths. The third component is driven by Panicoideae, leaf/husk and weight percent.

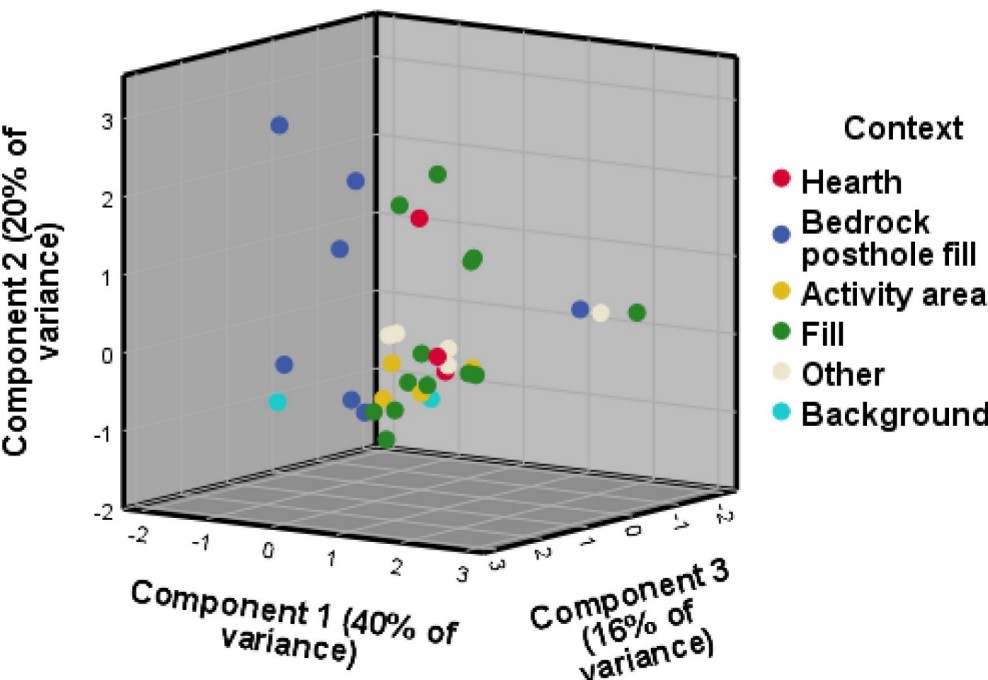

**Fig 7. PCA biplot for the phytolith results, WJ13.** The first component is driven by the variables monocots, leaf and leaf/stem, the second is negatively driven by dicots and single-cell phytoliths. The third component is driven by number of phytoliths per gram and multi-cell phytoliths.

elements measured in the sample, or the properties of the phytoliths, were examined to create a series of decisions according to the splitting variables (e.g. > or < 2079 PPM of phosphorus in the sample) chosen by the algorithm to best classify the maximum number of samples as per the excavators initial judgement (i.e. the identification of context in the field). The analysis provided a classification tree which can be visualised, and reported the amount of cases which were 'correctly' and 'incorrectly' classified according to the set parameters, i.e. samples which resemble the general trends observed within their context category.

The Weka decision trees created for the geochemistry results show a reliance on P, Si, Fe, Ca and S in distinguishing the clusters of the context categories for both sites (Figs 8 and 10). As with the PCA results, the decision trees created for WJ7 were more successful in clustering the data according to the context categories than the ones produced for WJ13, reflecting a clearer splitting process and higher percentage of correctly classified cases (Figs 8–10).

When plotting decision trees based on the phytolith analysis results, similar branching complexities can be seen as the ones created for the results of the geochemical analysis. The decision tree created for WJ7 had a higher amount of correctly classified cases than the one created for WJ13 (46%) and produced the purest divisions; Panicoideae and diatoms were used to differentiate between the background, deposit and posthole categories (Fig 10). The decision tree created for the phytolith results of WJ13 (Fig 9) stands in sharp contrast to it, with only 21% of cases correctly classified and extensive splitting required to produce nodes.

The lower amount of correctly classified cases and high degree of branching in the decision trees of WJ13 suggest that the context categories used in the analysis, or the division of samples to these, does not correspond well with the phytolith and geochemical data for this site. The high degree of branching also demonstrates why the J48 classifier was preferred in this analysis, as it produces a well pruned tree–other methods would create more complex trees.

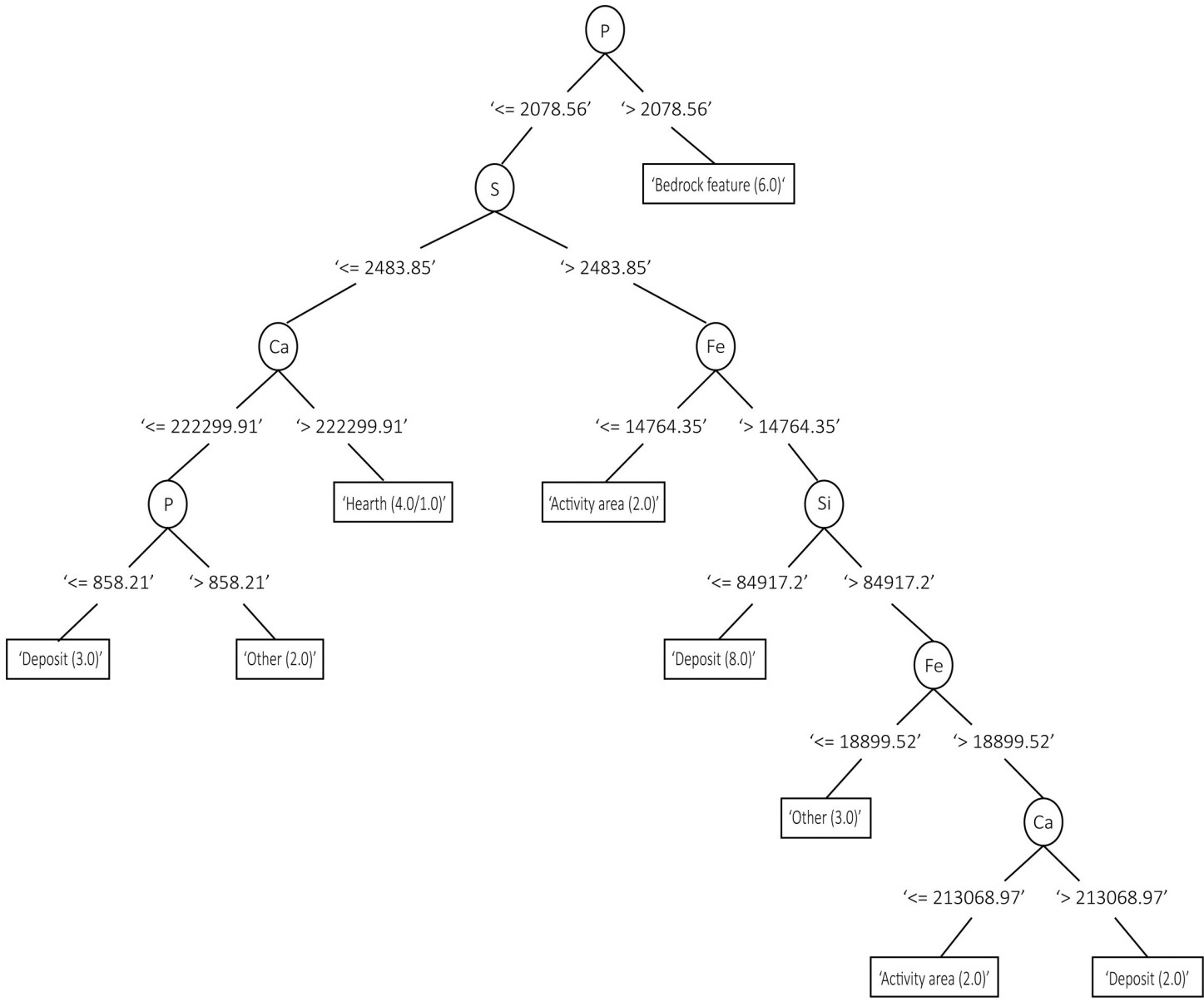

**Fig 8. Decision tree created for WJ13 based on the geochemical results, including the categories: Deposit, hearth, bedrock feature, activity area, fill and background.** 38% of cases were correctly classified. The numbers within each subset (or tree node) represent the amount of instances that are found within the subset. In cases where two numbers appear within the tree node, the first number indicates the 'correct' instances and the second reflects the 'incorrect' instances falling within the subset (i.e. samples having categories which agree or disagree with the category represented in the node). The numbers appearing between the tree nodes and the variables represent the splitting point, i.e. the value that split the instances according to those containing values of this variable that are smaller, larger or are equal to this number.

## Bayesian calculation

The model applied to the data combines information from the excavator's classification, the geochemistry and the phytolith analysis to provide an overall estimate of certainty in the classification of an activity area. Essentially, where both geochemistry and phytolith analysis results agree with the excavator's classification, the certainty of the excavator's classification increases. Where there is disagreement, if certainty falls below the prior probability (set according to excavator's classification), then the classification is reconsidered (Fig 11).

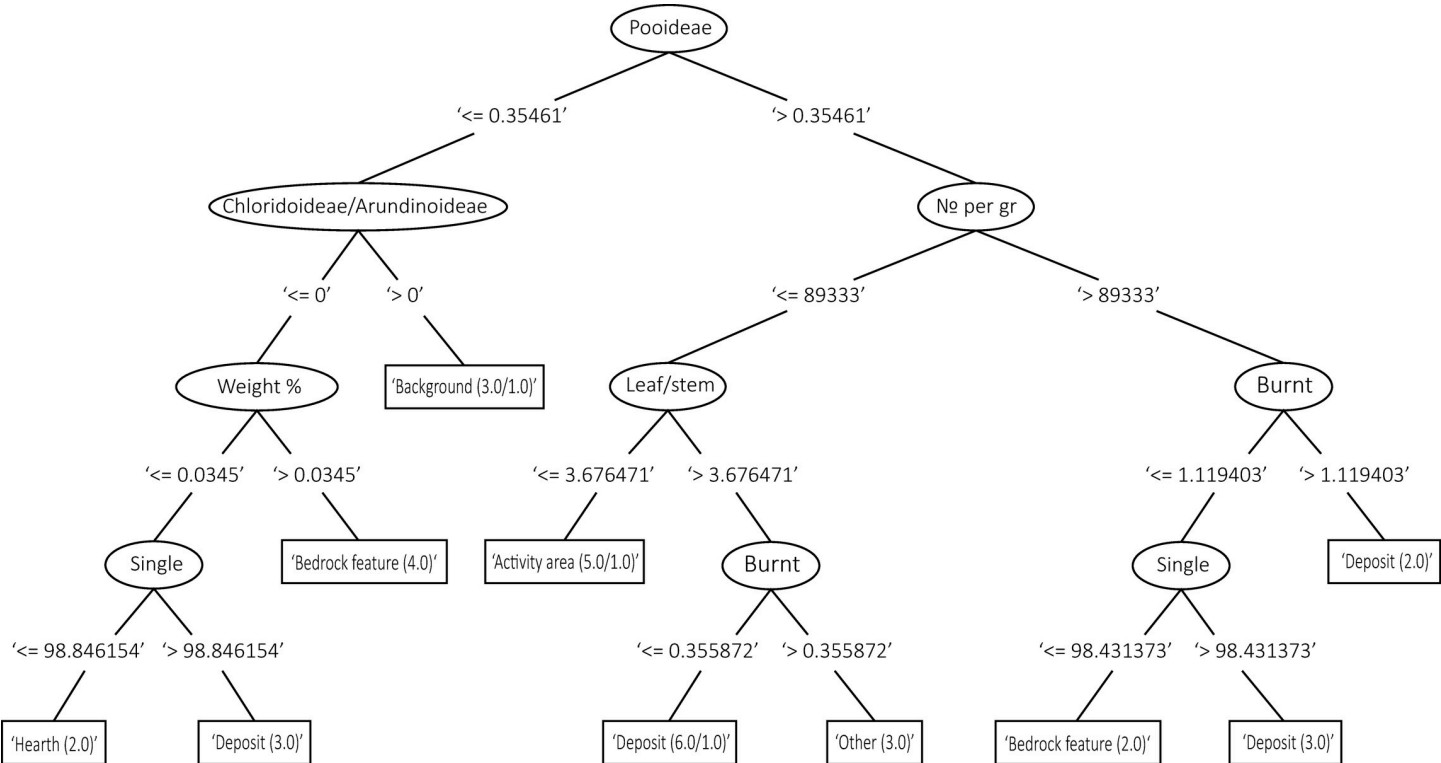

**Fig 9. Decision tree created for WJ13 based on the phytolith results, including the categories: Deposit, hearth, bedrock feature, activity area, fill and background.**
21% of cases were correctly classified. The numbers within each subset (or tree node) represent the amount of instances that are found within the subset. In cases where two numbers appear within the tree node, the first number indicates the 'correct' instances and the second reflects the 'incorrect' instances falling within the subset (i.e. samples having categories which agree or disagree with the category represented in the node). The numbers appearing between the tree nodes and the variables represent the splitting point, i.e. the value that split the instances according to those containing values of this variable that are smaller, larger or are equal to this number.

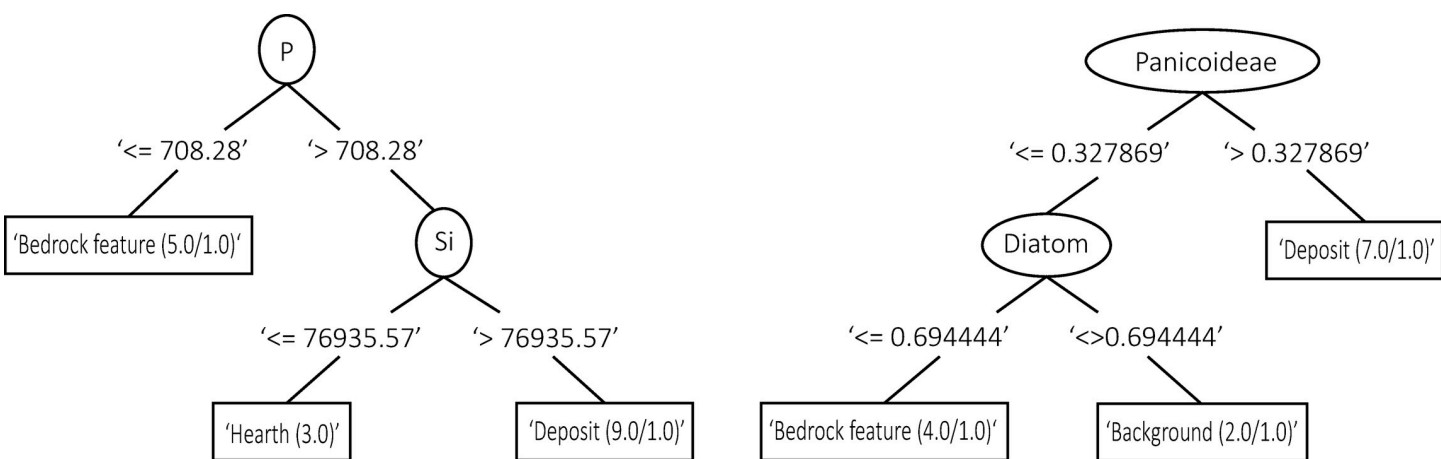

**Fig 10. Decision trees created for WJ7 based on the geochemical analysis (left) and phytolith counts (59% and 46% of cases correctly classified, respectively).** The numbers within each subset (or tree node) represent the amount of instances that are found within the subset. In cases where two numbers appear within the tree node, the first number indicates the 'correct' instances and the second reflects the 'incorrect' instances falling within the subset (i.e. samples having categories which agree or disagree with the category represented in the node). The numbers appearing between the tree nodes and the variables represent the splitting point, i.e. the value that split the instances according to those containing values of this variable that are smaller, larger or are equal to this number.

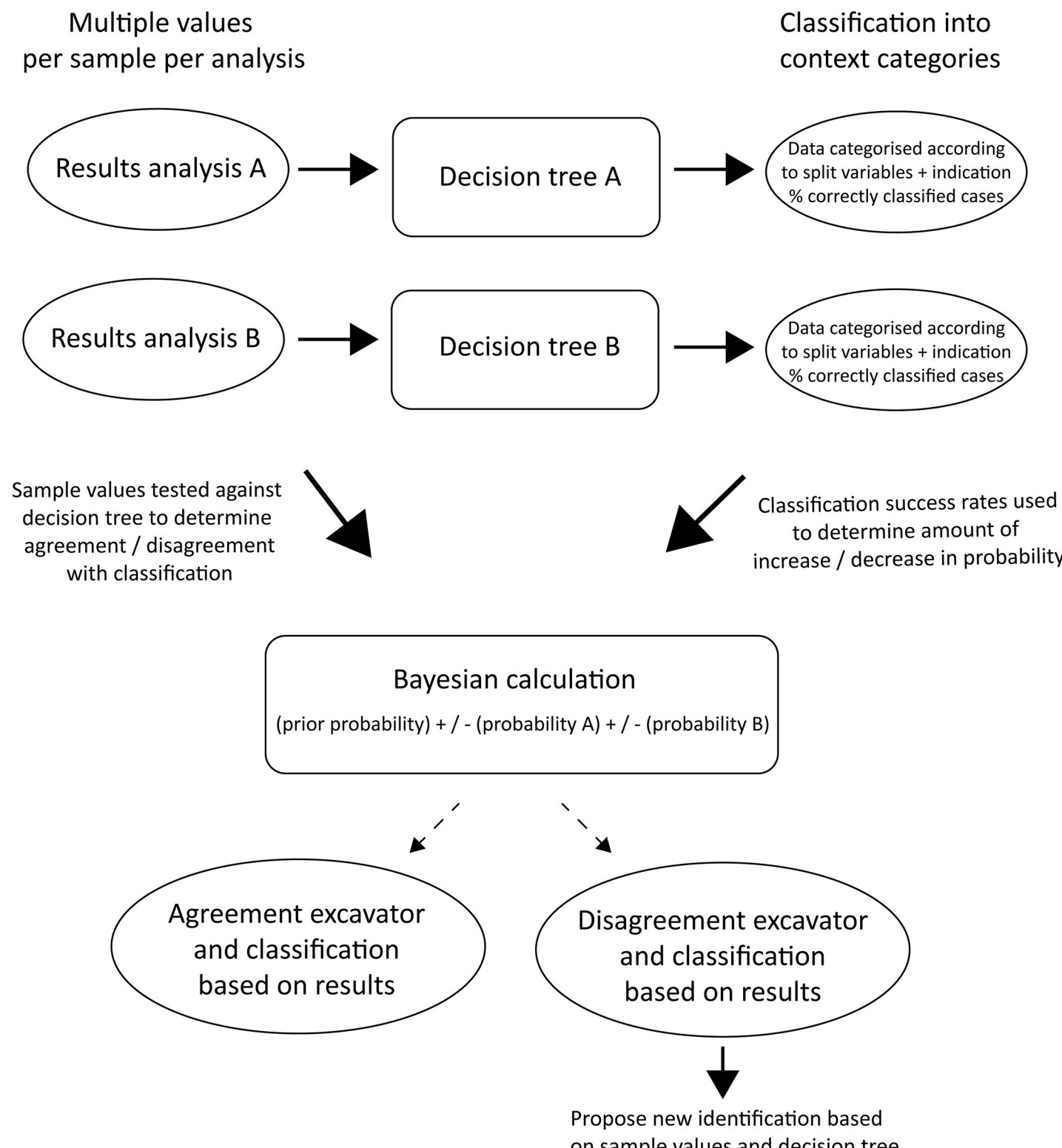

**Fig 11. Flowchart illustrating the use of decision trees and Bayesian calculation to combine the results of the soil analyses.**

Within the application of the model based on Bayesian confirmation in this study, the prior probability reflects expert opinion. It was set at 0.5, providing an 'uniformed prior'. The focus of this study was the addition of evidence from the geoarchaeological sources of information, and no attempt was made at estimating the certainty of the initial identification in the field. However, this input may vary in future studies. It may be that the expert opinion is given higher or lower weight, or perhaps that certain context categories, which are easier to identify, receive a higher prior probability while other categories, which are difficult to distinguish in the field, are attributed a lower prior probability (for example 0.7, or 0.4). Future studies might therefore choose to incorporate different prior probabilities for certain types of context categories within the model, based on the expected certainty of their initial identification.

The following equation was used:

$$P_{post} = P_{exc} + \{[(1 - P_{exc}) * \{a(P_{exc} * P_{geo}) + b(P_{exc} * P_{phyt})\}]/n\},$$

where $P_{post}$ is the final certainty of the classification, $P_{exc}$ is the excavator's certainty (the prior probability, set at 0.5 in this study), $P_{geo}$ and $P_{phyt}$ are the probabilities associated with overall correct classification from the decision trees (see description of results in previous section) and a and b are set to 1 when there is an agreement between excavator and decision tree classification and -1 where there is disagreement. n is the number of extra pieces of evidence used beyond excavator's classification, in this case 2. This approach has previously been successfully used in environmental studies to apply a Bayesian statistical framework to remaining uncertainty from a prior evaluation of probability, or to decrease certainty of a prior evaluation if new evidence disagrees [50, 51].

The equation was used as an excel function to calculate the probability of a correct identification of activity area in the field for each of the WJ13 samples. This was achieved by manually examining the values of the relevant splitting variable for both the geochemical and phytolith results based decision trees for each sample. Table 3 contains a list of the results of the application of the model to the samples from WJ13. In cases where there was a disagreement between the results of either the geoarchaeological or phytolith analyses, or both, an alternative context category was sought in line with the decision trees. This resulted in the reclassification of certain samples (Table 3).

The PCA 3D biplots below (Fig 12) visually illustrate the change in definition of activity for some of the samples following the reclassification of the results. Three samples were re-categorized as hearths, two as activity areas, and two posthole samples were reclassified as deposits. The new biplot portrays clearer clusters of context categories, especially when it comes to the hearth context category. The category 'other' now appears to share the same characteristics with the latter, and some of the features classified under this category, such as bins and pits (Table 1), might therefore reflect hearth locations which were not identified as such in the field.

The improved clustering of samples shown in the PCA scatterplot is not surprising, as the reclassification was based on the soil analysis results incorporated into the model. The increase of clustering was not the goal of this calculation. The PCA biplots merely demonstrate the changes in classification of the samples according to the geochemical and phytolith analyses results. In this way, the soil analyses can be combined and used as an additional indication or identification of archaeological features, guiding the archaeological interpretation of activity areas after excavation.

To illustrate how the reclassification of samples would change the interpretation of the use of space at the site, the location of the samples analysed in this study are presented with their associated context category before and after the application of the Bayesian model. Within the

**Table 3. Overview of the classification of samples from WJ13 in the field, the results of the application of the probability model, and proposed reclassification based on the results of the geochemical and phytolith analyses.** The prior probability was set at 0.5, the weight of the results of the geochemical and phytolith analyses was set at 0.38 and 0.21 (respectively) to reflect the amount of correctly classified cases in the decisions trees.

| Sample | Context | Prior probability | Geochemical results weight | Phytolith results weight | Both agree | Geochemical results disagree | Phytolith results disagree | Neither agree | Alternative category* |
|---|---|---|---|---|---|---|---|---|---|
| WJ13 5a 3 | Deposit | 0.5 | 0.38 | 0.21 | | 0.47875 | | | Other |
| WJ13 7a 5 | Deposit | 0.5 | 0.38 | 0.21 | | | | 0.42625 | Hearth |
| WJ13 8 8 | Deposit | 0.5 | 0.38 | 0.21 | | 0.47875 | | | Other |
| WJ13 16a 13 | Deposit | 0.5 | 0.38 | 0.21 | 0.57375 | | | | |
| WJ13 20b | Deposit | 0.5 | 0.38 | 0.21 | | | | 0.42625 | Other |
| WJ13 25 19 | Deposit | 0.5 | 0.38 | 0.21 | | | | 0.42625 | Deposit 2 |
| WJ13 50a 30 | Deposit | 0.5 | 0.38 | 0.21 | 0.57375 | | | | |
| WJ13 53a 32 | Deposit | 0.5 | 0.38 | 0.21 | 0.57375 | | | | |
| WJ13 56b 40 | Deposit | 0.5 | 0.38 | 0.21 | 0.57375 | | | | |
| WJ13 62a 40 | Deposit | 0.5 | 0.38 | 0.21 | 0.57375 | | | | |
| WJ13 70a 38 | Deposit | 0.5 | 0.38 | 0.21 | 0.57375 | | | | |
| WJ13 71b 80 | Deposit | 0.5 | 0.38 | 0.21 | 0.57375 | | | | |
| WJ13 83a 46 | Deposit | 0.5 | 0.38 | 0.21 | | | 0.52125 | | Deposit 2/hearth |
| WJ13 10a 9 | Other | 0.5 | 0.38 | 0.21 | | 0.47875 | | | Hearth |
| WJ13 22 17 | Other | 0.5 | 0.38 | 0.21 | | | 0.52125 | | Deposit |
| WJ13 47 29 | Other | 0.5 | 0.38 | 0.21 | | 0.47875 | | | Hearth |
| WJ13 52a 31 | Other | 0.5 | 0.38 | 0.21 | 0.57375 | | | | |
| WJ13 57a 33 | Other | 0.5 | 0.38 | 0.21 | | | 0.52125 | | Activity area |
| WJ13 15a 12 | Activity area | 0.5 | 0.38 | 0.21 | 0.57375 | | | | |
| WJ13 45a 46 | Activity area | 0.5 | 0.38 | 0.21 | 0.57375 | | | | |
| WJ13 59a 31 | Activity area | 0.5 | 0.38 | 0.21 | 0.57375 | | | | |
| WJ13 66b 39 | Activity area | 0.5 | 0.38 | 0.21 | 0.57375 | | | | |
| WJ13 12 12 | Hearth | 0.5 | 0.38 | 0.21 | 0.57375 | | | | |
| WJ13 18 13 | Hearth | 0.5 | 0.38 | 0.21 | 0.57375 | | | | |
| WJ13 22 14 | Hearth | 0.5 | 0.38 | 0.21 | | | 0.52125 | | Deposit |
| WJ13 24 20 | Posthole | 0.5 | 0.38 | 0.21 | | | 0.52125 | | Deposit/bedrock 2 |

(*Continued*)

**Table 3.** (Continued)

| Sample | Context | Prior probability | Geochemical results weight | Phytolith results weight | Both agree | Geochemical results disagree | Phytolith results disagree | Neither agree | Alternative category* |
|--------|---------|------------------|---------------------------|-------------------------|-----------|------------------------------|----------------------------|--------------|----------------------|
| WJ13 85 54 | Posthole | 0.5 | 0.38 | 0.21 | 0.57375 | | | | |
| WJ13 90a 56 | Posthole | 0.5 | 0.38 | 0.21 | 0.57375 | | | | |
| WJ13 92a 57 | Posthole | 0.5 | 0.38 | 0.21 | | | 0.52125 | | Deposit/bedrock 2 |
| WJ13 96 59 | Posthole | 0.5 | 0.38 | 0.21 | 0.57375 | | | | |
| WJ13 104 65 | Posthole | 0.5 | 0.38 | 0.21 | 0.57375 | | | | |

* The number 2 indicates cases where the suggested reclassification falls under the same context category according to the decision tree, but within a different node. In cases where the geochemical and phytolith results suggest different alternative categories, two reclassification options are provided (respectively).

plan of WJ13 showing the early and middle phases of occupation, the change is reflected in a broadening of a section of 'activity area' at the north-eastern end of the building, and two features at the south-western end of it now fall under the category 'other' (mainly representing pits and bins) instead of 'deposit' (Fig 13). The changes in the late occupation phase mainly concern the addition of external hearths to the building instead of the categories 'deposit' and 'other' (Fig 14).

## Discussion

This paper explored the use of a digital framework consisting of classification algorithms and Bayesian confirmation to integrate output from diverse analysis methods and sources of information. The principal behind this methodology is to translate the data into a form where they can be brought to the same level, so that information can be extracted that either supports or contradicts a particular hypothesis.

The case study presented here illustrates how such a framework can help maximise the information gained through analysis techniques when dealing with ambiguity in their interpretation. Although WJ13 and WJ7 share a similar environmental and historical setting, and are adjacent to one another, the results of the geochemical and phytolith analyses were not as straightforward in interpreting the use of space at WJ13 as with WJ7. The data produced for the latter site exhibits a clear classification into distinguishable context categories when examined through PCA—mainly due to the geochemical input, while the geochemical and phytolith analysis of WJ13 hints towards very subtle spatial trends and produced less straightforward PCA biplots. The decision trees created for the two sites tell the same story, those produced for WJ7 depict a compact tree and clear division into context categories, while the output trees for WJ13 portray extensive branching reflecting significantly lower classification success. It might be that the short-lived nature and relative simplicity of a structured occupation sequence of WJ7 contributed to the ease of its interpretation, while the relatively complex sequence of occupation at WJ13 made its interpretation less straightforward [15].

Because of the complexity, ambiguity and incompatibility of the geochemistry and phytolith analysis results for WJ13, additional means of data analysis were necessary for utilising these data to aid the interpretation of spatial trends at WJ13. Decision trees provided probabilities related to the success of classification into the pre-defined context categories, indicated how well these describe the data, and identified the key variables that split the data into context

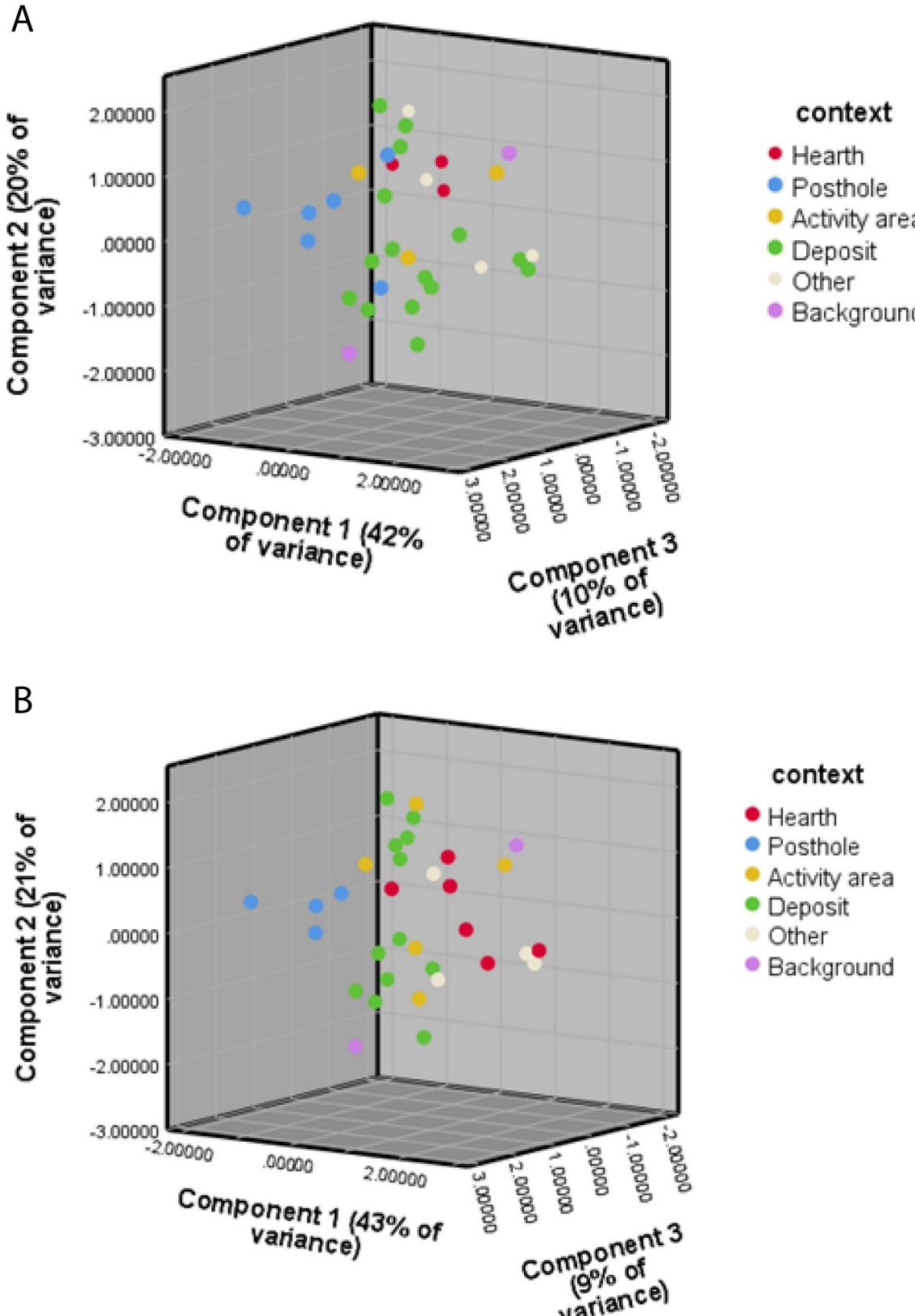

**Fig 12.** A) PCA biplot for WJ13 based on the geochemical analysis results; B) PCA biplot for WJ13 based on the geochemical analysis results after the change in the categories of some of the samples after the application of the Bayesian based model.

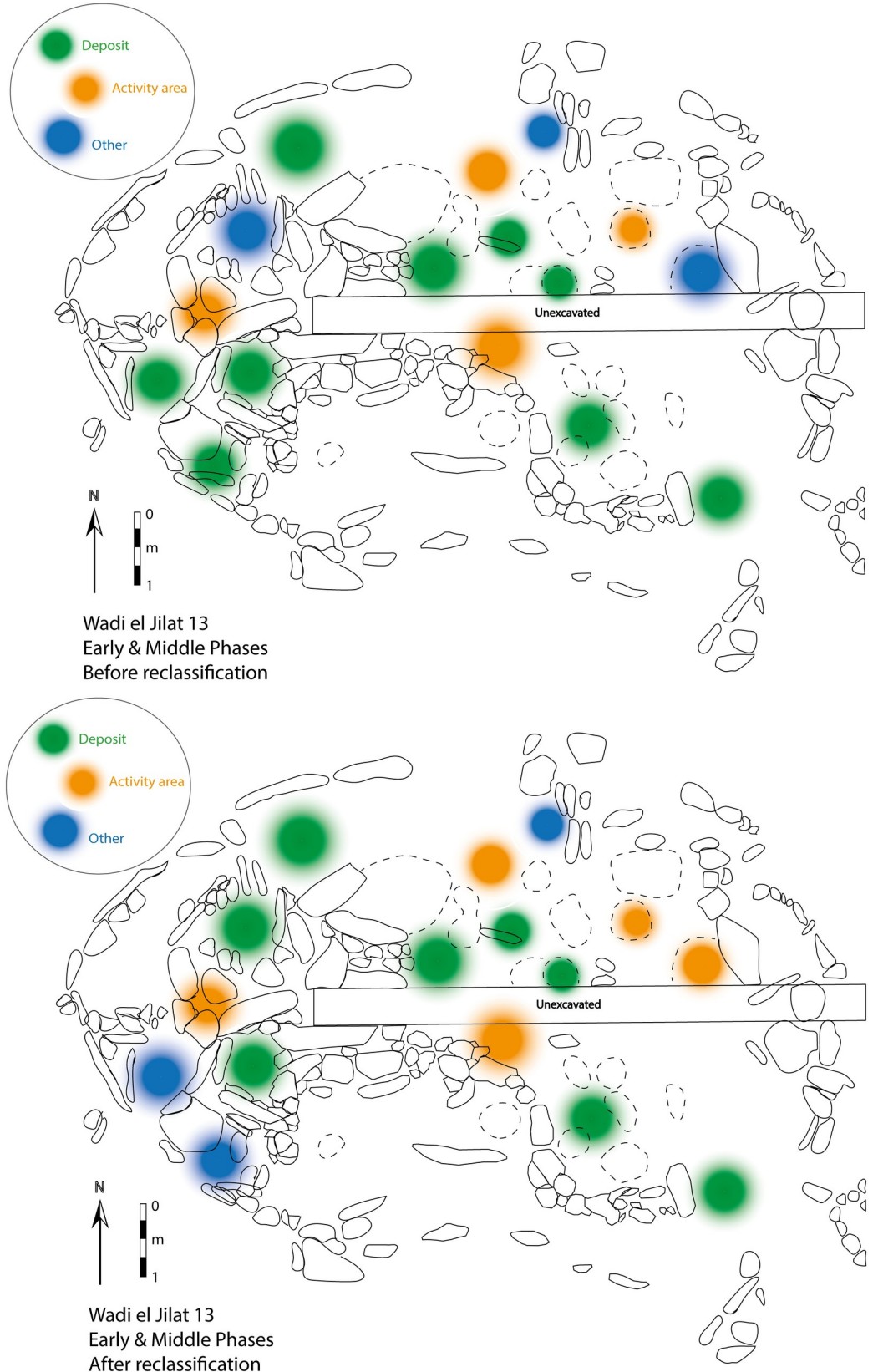

**Fig 13. A plan of early and middle phases at Wadi el-Jilat 13, before (top) and after (bottom) the reclassification of samples.**

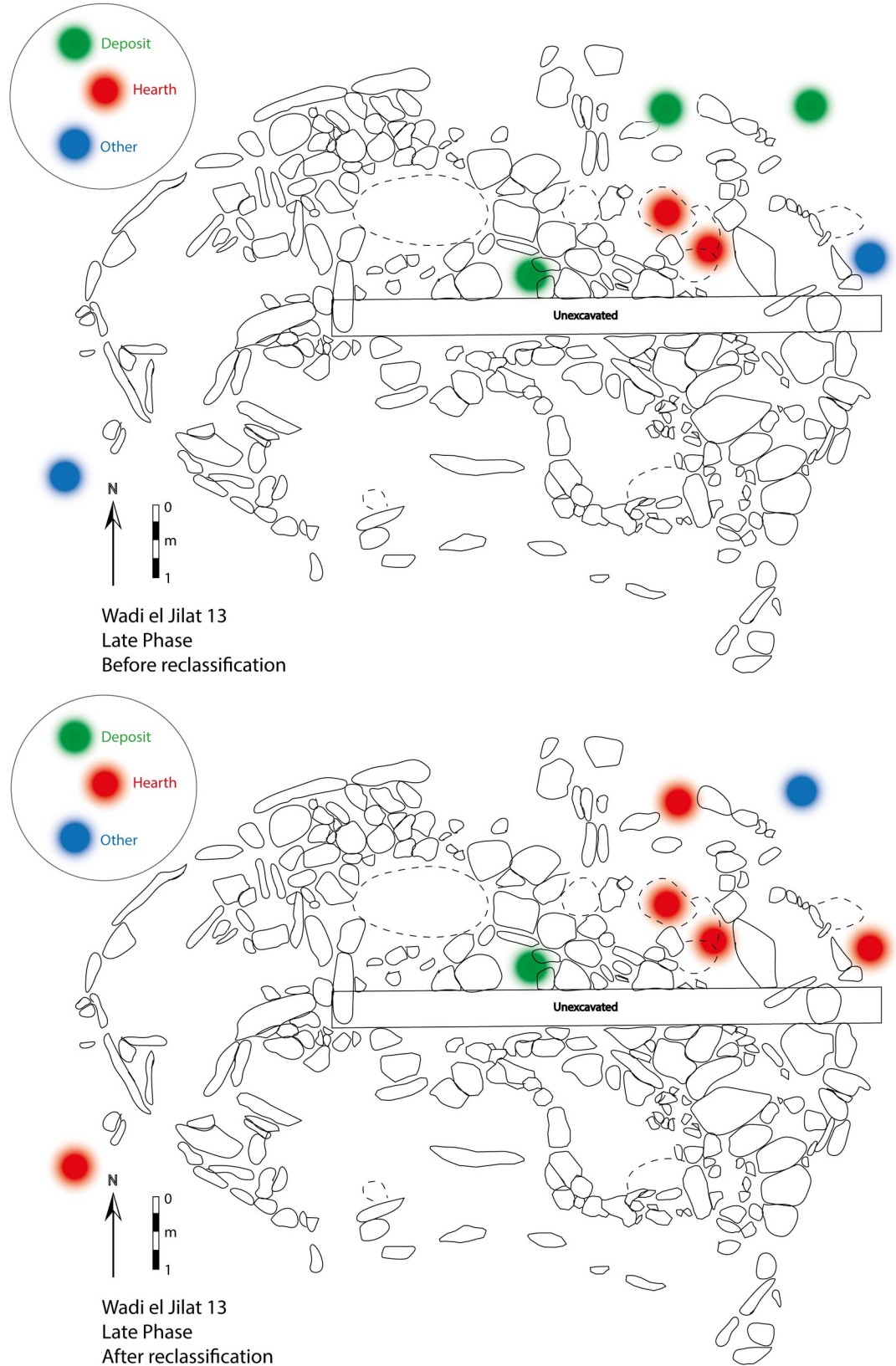

**Fig 14. A plan of late phase at Wadi el-Jilat 13, before (top) and after (bottom) the reclassification of samples.**

categories. This overview enabled the identification of typical signatures for specific activity areas, and made the division of each sample into a context category based on the results of the soil analysis possible.

The model based on Bayesian confirmation relied on this information in order to test and reclassify specific samples into the context group they resembled most by considering multiple sources of evidence. It could potentially incorporate any number of additional results to complement those of the geochemical and phytolith analyses. Several samples seemed to fit well within the pre-defined context category allocated to them, while others were reclassified according to the decision trees. While the process of testing which category each sample fits best was carried out manually in this study, it may be automated in future studies if necessary, for example when a large number of samples or sources of information is considered.

This manner of combining information from various sources of information within a single model, where they are considered independently from each other, carries much potential for aiding interpretation of ambiguous features in general. The application of this model to the samples from WJ13 illustrates that the use of even one type of additional evidence may improve the original interpretation of the use of space at a site, but that the certainty of the new identification increases when another method is added. In this sense, the difference in the type of data achieved from the two analysis techniques makes the identification of activity areas more convincing. The phytolith analysis, for example, reflects patterns of plant use, while the geochemistry is related to various signals of activities such as burning and animal husbandry. If both of these different sources point towards a confirmation or rejection of the initial interpretation of a context category, it is more compelling than is the case with more closely related sources of information (such as phytoliths and macrobotanical remains for example).

Overall, the reclassification of the WJ13 samples resulted in a reduction of the more common, general context categories 'deposit' and 'other' in favour of more specific context categories such as the 'hearth' and 'activity area' categories. It could be that these context categories are more difficult to identify in the field in some cases, while the phytolith and geochemical traces in the soil provide additional microscopic means to distinguish them. On the other hand, it is not possible to determine which interpretation is more accurate; the original identification in the field or the one based on the results of the phytolith and geochemical analyses, since the reclassification according to the model merely reflects the latter. The model refines the initial interpretation of activity areas by using the analyses results as either supporting or contradicting evidence. The 'correct' level of classification was low in our case, which means that the geoarchaeological results did not have a large effect on the final probabilities. Had the decision trees predicted 90% of cases correctly, they would have had a greater influence on the final probability. Weak classification success would result in a greater weighting on the classifier's results, and vice versa.

Nevertheless, it is encouraging to observe that when plotted against the plan of the site, the reclassification of samples does not seem to be spatially random. Clearer spatial clusters of the categories 'activity area', 'hearth' and 'other' were formed after the change. This suggests that the model was successful in identifying and applying valid indicators of activity areas in the geochemistry and phytolith data, which appear to provide a meaningful alternative interpretation of space at the site. Future studies could help establish the value of the approach presented in this paper for interpreting different types of sites and identify other applications–such as testing contradicting sources of evidence, or its use for archaeological prospection.

## Conclusions

While increasingly used to aid archaeological interpretation, scientific analysis techniques often generate results which are equivocal, subtle or distinct, and therefore difficult to relate

back to past human behaviour. A multiproxy approach can help combat issues of equifinality and ambiguity, but produces incompatible results due to differences in the level of measurement of various sources of information.

By exploring a new way to combine multiproxy data, this research aimed to maximise the information gained from sites which are difficult to interpret. Rather than trying to find "hard" archaeological evidence, the approach taken in this study was to bridge the gap between the scientific methods used and the ambiguity inherent in archaeological data. This required fitting, or scaling down, hard methods to soft data, which was enabled by the use of decision trees and Bayesian inference. By allowing for expert opinion to contribute to the outcome of the model, and dealing with the uncertainty typical to archaeological data, such models may provide a useful tool for the incorporation of the wide range of sources of information that archaeologists must consider while interpreting ancient human behaviour. The potential applications of this model are broader than archaeology, as any field seeking to incorporate expert opinion with additional sources of information could benefit from its application.

## Supporting information

**S1 File. Geochemical and phytolith data for WJ7 and WJ13 relied on for the analysis in this paper, in a format used as input for the WEKA decision trees.**
(CSV)

## Acknowledgments

The dataset used as input for the analysis in this paper has been provided as a S1 File. The geochemistry and phytolith raw data can be accessed through the Bournemouth University website: http://eprints.bournemouth.ac.uk/29485/. We would like to thank Timothy Darvill and Kate Welham for their involvement in this project, which included much valuable guidance and advice.

No permits were required for the described study, which complied with all relevant regulations.

## Author Contributions

**Conceptualization:** Daniella Vos, Emma L. Jenkins.

**Data curation:** Daniella Vos.

**Formal analysis:** Daniella Vos, Richard Stafford.

**Funding acquisition:** Emma L. Jenkins.

**Investigation:** Daniella Vos.

**Methodology:** Daniella Vos, Richard Stafford.

**Project administration:** Daniella Vos.

**Resources:** Emma L. Jenkins, Andrew Garrard.

**Software:** Richard Stafford.

**Supervision:** Emma L. Jenkins.

**Validation:** Richard Stafford.

**Visualization:** Daniella Vos.

**Writing – original draft:** Daniella Vos.

**Writing – review & editing:** Daniella Vos, Richard Stafford, Emma L. Jenkins, Andrew Garrard.

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
