## [Decision Letter · Decision Letter 0]

14 Sep 2020

PONE-D-20-17716

A model based on Bayesian inference and machine learning algorithms to aid archaeological interpretation by integrating incompatible data

PLOS ONE

Dear Dr. Vos,

Thank you for submitting your manuscript to PLOS ONE. After careful consideration, we feel that it has merit but does not fully meet PLOS ONE’s publication criteria as it currently stands. Therefore, we invite you to submit a revised version of the manuscript that addresses the points raised during the review process.

We look forward to receiving your revised manuscript.

Kind regards,

Ceren Kabukcu, PhD

Academic Editor

PLOS ONE

Journal Requirements:

2.In your manuscript, please provide additional information regarding the specimens used in your study. Ensure that you have reported specimen numbers and complete repository information, including museum name and geographic location.

For more information on PLOS ONE's requirements for paleontology and archaeology research, see https://journals.plos.org/plosone/s/submission-guidelines#loc-paleontology-and-archaeology-research.

3.We note that [Figure(s) 1] in your submission contain [map/satellite] images which may be copyrighted. All PLOS content is published under the Creative Commons Attribution License (CC BY 4.0), which means that the manuscript, images, and Supporting Information files will be freely available online, and any third party is permitted to access, download, copy, distribute, and use these materials in any way, even commercially, with proper attribution. For these reasons, we cannot publish previously copyrighted maps or satellite images created using proprietary data, such as Google software (Google Maps, Street View, and Earth). For more information, see our copyright guidelines: http://journals.plos.org/plosone/s/licenses-and-copyright.

1.    You may seek permission from the original copyright holder of Figure(s) [1] to publish the content specifically under the CC BY 4.0 license. 

Additional Editor Comments (if provided):

Dear authors,

Many apologies for a lengthy peer-review and decision process. The reviews, as you will see below, are polarised. Reviewer 1 has suggested an outright rejection of the manuscript, while Reviewer 2 suggests Minor Revision. Having read through both sets of comments, the manuscript and made an evaluation, I am recommending major improvements to the manuscript in order to proceed with further peer review. In your resubmission, please address all points raised by both reviewers.

As highlighted by Reviewer 1, there are serious concerns with a thorough description and theoretical/methodological backing with regard to the use of Bayesian statistics in addressing the research questions raised in the manuscript. This point is also raised by Reviewer 2, albeit with different wording. Namely, the issue of identifying and determining the contextual interpretation of specific archaeological features and/or occupation areas is presented in the manuscript as being better represented through geochemical & phytolith analysis supported by statistical examination of the resulting datasets. However, the question still remains as to which method (i.e., laboratory analysis of materials deriving from a site or contextual observations during excavation) are or should be considered more definitive. As suggested by Reviewer 2, this theoretical point must be clarified throughout the text, making explicit the fact that in order to justify the interpretations presented in the manuscript further work at archaeological sites with more definitive contextual observations could follow the presented research in order to be able to test some of the assumptions.

With regard to the details on statistical methodologies used, please address all points raised by reviewer 1:

“From the theory presented in the paper (line 269), it seems that Bayes theorem may have been used to combine conditional probabilities. However, it is unclear why this specific equation is used. They do justify the prior probability, albeit later in the paper (lines 414 - 415). However, it is not clear how the rest of the equation on line 269 is derived. For a full inference approach, we would expect to see a clear statement of a likelihood or at least a series of conditional probability statements, but these are not present. Without formality of this sort, use of the words “Bayesian inference” and “model” in the title and text (lines 81, 83, 86, 89, 122, 264, 334, 338) are misleading.”

In light of these comments, I would suggest either a full-scale re-writing of the methodological aspects of the article, removing statements pertaining to Bayesian inference and modelling. Alternatively, you could provide a full and detailed rebuttal of the points raised by Reviewer 1, or could seek to modify your application of this aspect of the manuscript.

With regard to the application of PCA & multivariate analyses, please address all points raised by both reviewers. More specifically, please also provide further examples from already published studies as highlighted by Reviewer 2 (i.e., Rondelli et al., Lancelloti et al. and Peto et al.). With regard to the discussion of PCA and presentation of the results, I encourage the authors to also include a fuller justification of the choice of multivariate technique to be used on the datasets (see Reviewer 1 comments). For example, other methods such as Multiple Factor Analysis or other multivariate techniques enabling the analysis of datasets of mixed sources/scales of recording and/or different natures were not employed in the present study. Alternatively, if the aim of the study is to examine a characterisation/discrimination of context types with regard to the data categories included, other techniques such as cluster analysis or discriminant functions could have also been employed. Furthermore, I suggest that the authors also present a more solid justification with regard to the choice of 3D plotting, and provide the results of PCA in full as supplementary materials (e.g., contributions to dimensions, inertia etc).

As a final comment, while the text reads clear and concise, I ask that the authors revise certain statements throughout the text which refer to the nature of archaeological data as ambiguous or carrying an inherent ambiguity or vagueness (particularly with regard to contextual data collection during excavation). There is no need to view archaeological datasets, however fragmentary or incomplete, as any less rigorous and/or representative.

Kind Regards,

Ceren Kabukcu

Reviewers' comments:

Reviewer's Responses to Questions

**Comments to the Author**

1. Is the manuscript technically sound, and do the data support the conclusions?

Reviewer #1: No

Reviewer #2: Yes

2. Has the statistical analysis been performed appropriately and rigorously? 

Reviewer #1: No

Reviewer #2: Yes

3. Have the authors made all data underlying the findings in their manuscript fully available?

Reviewer #1: Yes

Reviewer #2: Yes

4. Is the manuscript presented in an intelligible fashion and written in standard English?

Reviewer #1: Yes

Reviewer #2: Yes

5. Review Comments to the Author

Reviewer #1: The authors of this paper suggest, in the title and abstract, that they are offering a novel, Bayesian, model-based approach for combining partial or incomplete archaeological data, thus improving the quality of the archaeological interpretation from a given site. However, what is presented in the main body of the paper is a collection of ad hoc tools with limited statistical or methodological motivation or justification. Here we outline four main concerns: lack of methodological motivation for the choice of methods; lack of discussion of other methods that could have been used; uncertainty as to whether Bayesian inference has actually been carried out; and weak motivation for and details of the statistical theory that is used.

The authors offer archaeological motivation of the need for methods to combine formally information from different sources, but they do not justify their selection of the particular suite of methods chosen and offer no evidence as to what other methods were considered. The authors do present some reasoning for using Bayesian inference (lines 121-125). However, this does not justify why using Bayesian inference in conjunction with classification trees is the most suitable method for combining related sources of rather limited archaeological data, of the sort described.

Furthermore, it is unclear from the text whether Bayesian inference has actually been carried out. From the theory presented in the paper (line 269), it seems that Bayes theorem may have been used to combine conditional probabilities. However, it is unclear why this specific equation is used. They do justify the prior probability, albeit later in the paper (lines 414 - 415). However, it is not clear how the rest of the equation on line 269 is derived. For a full inference approach, we would expect to see a clear statement of a likelihood or at least a series of conditional probability statements, but these are not present. Without formality of this sort, use of the words “Bayesian inference” and “model” in the title and text (lines 81, 83, 86, 89, 122, 264, 334, 338) are misleading.

Aside from the ambiguity regarding the use of Bayesian inference, it is also unclear why PCA is used in this paper. One might say that it was included as an exploratory data analysis tool, to give the reader a sense of the data. However, this has not been motivated and so its appearance (at line 158) is rather abrupt. Furthermore, later the authors appear initially to be using PCA to justify the success of their method (by showing improved clustering), which is misleading since they use the same data for both the classification trees and the PCA. The authors do address this multiple use of the same data in the subsequent paragraph (lines 363-369), but including this plot nonetheless suggests some circularity in the author's thinking.

Specific comments that we hope might help in any revisions to the text:

Throughout: Abbreviations such as PCA and XRF are not defined.

58: It would be helpful to reference some standard textbooks for readers unfamiliar with phtolyith analysis, micromorphology, or lipid residue analysis.

65: It is unclear what is meant by the phrase “higher interpretive power”. Since power has a technical statistical meaning, this may mislead the reader.

90: Not sure about the use of the word vagueness, what does this mean in the context of the data?

257-261: The meaning of this section is very unclear, but is vital to the value or otherwise of the work described. In order to fully understand and verify the methodology, considerably greater detail is needed here about the Weka software, the J48 classifier and the concepts of correctly and incorrectly classifying “according to set parameters”. Adoption of these tools also lacks any real motivation. Why these particular tools and not the many other similar ones available?

293-309: There seems to be a typo in one or other of these captions since they both refer to the same dataset (WJ13).

333-349 (and Table 3): The reviewers could not discern the methodology from the description offered here and, as a result, could not understand the contents of Table 3. Precisely what are the numbers in Table 3, how should we interpret them given the equation provided and why are so many of the numbers in each column identical?

Reviewer #2: The submitted manuscript presents a study on data integration within household archaeological research. The subject of the study is up-to-date, because the integration of data that have dissimilar dimensions is always an important and sometimes problematic part of multi-proxy approaches.

Lines 63-75: The circumstances of multi-proxy studies in general, which are described in this paragraph are true, however I would suggest to rephrase it a little, since there are existing ways - even if partly debatable - to perform data integration by the means of standardization and normalization. These preparatory tools in statistics can be utilized on data sets with different dimensions in order to integrate them within an interpretive multi elemental statistical approach such as PCA, Cluster and many others.

Attempts on this can be consulted in Rondelli et al. 2014 Anthropic activity markers and spatial variability: an ethnoarchaeological experiment in a domestic unit of northern Gujarat (India). J Archaeol Sci 41:482-492, or Lancelotti et al. Investigating fuel and fireplaces with a combination of phytoliths and multielement analysis; an ethnographic experiment VHA 26: 75-83, or Pető et al 2014: Activity area analysis of a Roman period semi-subterranean building by means of integrated archaeobotanical and geoarchaeological data VHA 24: 101-120.

In addition to the abovementioned this paper presents another and in this sense novel approach to data integration. It uses data from a previous study. Materials and methods, as well as data are partly presented and described in the introduction of the manuscript. The way the older data is presented is sufficient for the understanding of the new integration approach, however one must delve deep into the PhD dissertation of the corresponding author to get a more detailed picture.

Lines 152-153: Why is it interesting that K and Mg elevations occur in hearth areas? To my best knowledge ash contains a lot of these elements.

Lines 432-437: I think this is an evident statement throughout this profession therefore it doesn’t really need to be addressed here. This derives from the nature of multi-proxy studies that different elements of the ’whole mosaic’ can be reconstructed with the different pairing of the various approaches.

The main result of the paper is shown in Fig 13. The entire attempt focuses on the re-classification of the household contexts based on the integration of the phytolith and geochemical dataset. This is very important; however, the results do not appear plausible on this figure. It would be worth improving the visualization of this, because this is one of the key elements of the paper.

General comment: The paper attempts to show that the statistical way that was utilized is a proper way to aid the functional identification of archaeological features and contexts. The paper states that the case study was carried out on a site where the remains of the material culture are extremely sparse, therefore the identification of the archaeological features and contexts was revisited and certain modification could have been established. The concept of the paper is clear and the approach is fine, however how can the authors be sure that the re-classification of the features and contexts is valid if there is no archaeological evidence to support that? With other words, it would be great to repeat this study at a site which provides enough ‘classic’ archaeological evidence, so the blind test of the phytolith and geochemical data could be backed up and the method could be validated. This logic should be at least mentioned in the discussion so that readers understand the limitation of the approach.

6. PLOS authors have the option to publish the peer review history of their article (what does this mean?). If published, this will include your full peer review and any attached files.

Reviewer #1: No

Reviewer #2: No

---

## [Author Response · Author response to Decision Letter 0]

25 Nov 2020

Dear editor and reviewers, 

We would like to thank you for the time and effort you have taken to review our manuscript, and for your helpful comments which have enabled us to improve it. We respond to specific comments below. As for the general editorial comments, we have revised our text to refer more kindly to the nature of archaeological data. In addition, three figures which were subject to copyright have been replaced. 

Response to reviewers:

Reviewer comments marked with -, response text marked with >

Reviewer #1: 

- The authors of this paper suggest, in the title and abstract, that they are offering a novel, Bayesian, model-based approach for combining partial or incomplete archaeological data, thus improving the quality of the archaeological interpretation from a given site. However, what is presented in the main body of the paper is a collection of ad hoc tools with limited statistical or methodological motivation or justification. Here we outline four main concerns: lack of methodological motivation for the choice of methods; lack of discussion of other methods that could have been used; uncertainty as to whether Bayesian inference has actually been carried out; and weak motivation for and details of the statistical theory that is used.

> We would like to thank the reviewer for their comments, which have definitely helped us to improve our manuscript and better formulate our arguments. This has enabled us to present our work in a more comprehensive manner. We address the specific comments below.

- The authors offer archaeological motivation of the need for methods to combine formally information from different sources, but they do not justify their selection of the particular suite of methods chosen and offer no evidence as to what other methods were considered. The authors do present some reasoning for using Bayesian inference (lines 121-125). However, this does not justify why using Bayesian inference in conjunction with classification trees is the most suitable method for combining related sources of rather limited archaeological data, of the sort described.

> We agree that a proper justification is missing, and have added more information about our motivation to use Bayesian inference and decision trees in the introduction. 

We do not claim to have found the ultimate, most suitable method for combining related sources of data, but a highly suitable one that worked well and may therefore be useful to others. The justification is therefore limited to why this is the case.

- Furthermore, it is unclear from the text whether Bayesian inference has actually been carried out. From the theory presented in the paper (line 269), it seems that Bayes theorem may have been used to combine conditional probabilities. However, it is unclear why this specific equation is used. They do justify the prior probability, albeit later in the paper (lines 414 - 415). However, it is not clear how the rest of the equation on line 269 is derived. For a full inference approach, we would expect to see a clear statement of a likelihood or at least a series of conditional probability statements, but these are not present. Without formality of this sort, use of the words “Bayesian inference” and “model” in the title and text (lines 81, 83, 86, 89, 122, 264, 334, 338) are misleading.

> We agree that the description of the use of the Bayesian equation may seem unclear as certain elements in its explanation are provided later on in the manuscript. We hope that this has now been improved by some editing. The explanation of the prior probability has been moved to the ‘methods and results’ section. 

The title states that the model is based on Bayesian inference. Additional definition and explanation of how the model applied in this study relates to Bayesian inference has now been provided in the introduction. We also replaced the reference to Bayesian inference with the more specific term Bayesian confirmation to help avoid misunderstanding. In addition, the methodology section has been updated to include more detail and explanation, now including the results. We hope that the rationale and background to the equation are now clear.

- Aside from the ambiguity regarding the use of Bayesian inference, it is also unclear why PCA is used in this paper. One might say that it was included as an exploratory data analysis tool, to give the reader a sense of the data. However, this has not been motivated and so its appearance (at line 158) is rather abrupt. Furthermore, later the authors appear initially to be using PCA to justify the success of their method (by showing improved clustering), which is misleading since they use the same data for both the classification trees and the PCA. The authors do address this multiple use of the same data in the subsequent paragraph (lines 363-369), but including this plot nonetheless suggests some circularity in the author's thinking.

> The PCA is indeed used here as an explanatory analysis tool, to give the reader a sense of the data. The use of PCA could have been misleading if it were not for the accompanying text in lines 359-365: “The improved clustering of samples shown in the PCA scatterplot is not surprising, as the reclassification was based on the soil analysis results incorporated into the model. The increase of clustering was not the goal of this calculation. The PCA biplots merely demonstrate the changes in classification of the samples according to the geochemical and phytolith analyses results. In this way, the soil analyses can be combined and used as an additional indication or identification of archaeological features, guiding the archaeological interpretation of activity areas after excavation.” 

In other words, the aim of using the PCA biplots is not for data analysis purposes, but for visualisation purposes (this is also mentioned in lines 346-347). We are clearly stating here that we are not trying to justify the success of our method, but to illustrate the change after applying it.

If this clarification is seen to demonstrate awareness of this issue (“The authors do address this multiple use of the same data in the subsequent paragraph”), we do not understand how our thinking is still deemed to portray circularity in the matter. We feel that the clarification in lines 359-365 is sufficient in revealing our intention in illustrating the results by means of PCA biplots, and in addressing the issue of circularity.

- Specific comments that we hope might help in any revisions to the text:

Throughout: Abbreviations such as PCA and XRF are not defined.

Definitions for XRF and PCA have been added.

58: It would be helpful to reference some standard textbooks for readers unfamiliar with phtolyith analysis, micromorphology, or lipid residue analysis.

> Additional references have been added.

- 65: It is unclear what is meant by the phrase “higher interpretive power”. Since power has a technical statistical meaning, this may mislead the reader.

> The word power has been replaced with value.

- 90: Not sure about the use of the word vagueness, what does this mean in the context of the data?

> The word vagueness has been removed.

- 257-261: The meaning of this section is very unclear, but is vital to the value or otherwise of the work described. In order to fully understand and verify the methodology, considerably greater detail is needed here about the Weka software, the J48 classifier and the concepts of correctly and incorrectly classifying “according to set parameters”. Adoption of these tools also lacks any real motivation. Why these particular tools and not the many other similar ones available?

> This section has now been extended to include a better overview and rationale for the use of the WEKA J48 classifier, which was preferred here as it produces a well pruned tree. This leads to fewer possible choices, and therefore provides conservative probability values compared to others – which means that we will not over predict when combining these in the Bayesian model. 

- 293-309: There seems to be a typo in one or other of these captions since they both refer to the same dataset (WJ13).

> There is no typo. The captions in this section refer to three figures, one illustrating the decision tree created for the geochemical results for WJ13, one for the phytolith results for WJ13, and one illustrating two decision trees created for the geochemical and the phytolith results for WJ7 (in total 4 decision trees, those created for WJ7 were much smaller and could fit in a single figure).

- 333-349 (and Table 3): The reviewers could not discern the methodology from the description offered here and, as a result, could not understand the contents of Table 3. Precisely what are the numbers in Table 3, how should we interpret them given the equation provided and why are so many of the numbers in each column identical?

> We hope that the restructuring of the methodology section, now including the results, and the addition of an explanation in the caption of table 3, has made the contents of the latter clear.

Reviewer #2: 

- The submitted manuscript presents a study on data integration within household archaeological research. The subject of the study is up-to-date, because the integration of data that have dissimilar dimensions is always an important and sometimes problematic part of multi-proxy approaches.

> We would like to thank the reviewer for their useful and positive comments, which have helped us to improve our manuscript and present our work in a more comprehensive manner. 

- Lines 63-75: The circumstances of multi-proxy studies in general, which are described in this paragraph are true, however I would suggest to rephrase it a little, since there are existing ways - even if partly debatable - to perform data integration by the means of standardization and normalization. These preparatory tools in statistics can be utilized on data sets with different dimensions in order to integrate them within an interpretive multi elemental statistical approach such as PCA, Cluster and many others.

Attempts on this can be consulted in Rondelli et al. 2014 Anthropic activity markers and spatial variability: an ethnoarchaeological experiment in a domestic unit of northern Gujarat (India). J Archaeol Sci 41:482-492, or Lancelotti et al. Investigating fuel and fireplaces with a combination of phytoliths and multielement analysis; an ethnographic experiment VHA 26: 75-83, or Pető et al 2014: Activity area analysis of a Roman period semi-subterranean building by means of integrated archaeobotanical and geoarchaeological data VHA 24: 101-120.

> Thank you for directing our attention to these studies which indeed clearly illustrate the integration of the results of multiple techniques through standardization and normalization to enable their combined use in multivariate statistics. This is a valid point, and we have now addressed it in the text including the mentioned references. 

- In addition to the abovementioned this paper presents another and in this sense novel approach to data integration. It uses data from a previous study. Materials and methods, as well as data are partly presented and described in the introduction of the manuscript. The way the older data is presented is sufficient for the understanding of the new integration approach, however one must delve deep into the PhD dissertation of the corresponding author to get a more detailed picture.

> We tried to keep the description of the geoarchaeological analysis and results to a minimum since they are not the main focus of this paper, and are glad to hear that you find it sufficient for understanding the quantitative approach. To make it easier to find out more about the geoarchaeological results and their background, we have now added an additional reference of an (open access) publication in press in which these are presented in more detail: 

56. Vos D. New techniques for tracing ephemeral occupation in arid, dynamic environments: case studies from Wadi Faynan and Wadi al-Jilat, Jordan. In: Akkermans PMMG, editor. Landscapes of Survival. The Archaeology and Epigraphy of Jordan’s North-Eastern Desert and Beyond. Leiden: Sidestone Press; 2020.

- Lines 152-153: Why is it interesting that K and Mg elevations occur in hearth areas? To my best knowledge ash contains a lot of these elements.

> This sentence is confusing, and has been adjusted. It refers to the first sentence in the paragraph which states that there are no remarkable trends when comparing elements across context categories at the site of WJ7. In this respect, we would expect this site (WJ7) to show more differentiation then WJ13 since other means of analysis (PCA and decision trees) were able to portray clear divisions into activity areas. However, WJ13 shows clearer trends than WJ7 when average measurements are compared across contexts.

- Lines 432-437: I think this is an evident statement throughout this profession therefore it doesn’t really need to be addressed here. This derives from the nature of multi-proxy studies that different elements of the ’whole mosaic’ can be reconstructed with the different pairing of the various approaches.

> While this statement might be evident (for us), it is crucial for understanding the need for the approach presented in this paper. Since PLOS ONE caters to a broad audience, and this statement does not appear elsewhere in the text, we would prefer to keep it in order to make sure that this argument comes across even if the reader is not familiar with the rationale behind multi-proxy studies.

- The main result of the paper is shown in Fig 13. The entire attempt focuses on the re-classification of the household contexts based on the integration of the phytolith and geochemical dataset. This is very important; however, the results do not appear plausible on this figure. It would be worth improving the visualization of this, because this is one of the key elements of the paper.

> We have replaced figures 13 and 14, and hope that the new versions illustrates the change after reclassification more clearly.

- General comment: The paper attempts to show that the statistical way that was utilized is a proper way to aid the functional identification of archaeological features and contexts. The paper states that the case study was carried out on a site where the remains of the material culture are extremely sparse, therefore the identification of the archaeological features and contexts was revisited and certain modification could have been established. The concept of the paper is clear and the approach is fine, however how can the authors be sure that the re-classification of the features and contexts is valid if there is no archaeological evidence to support that? With other words, it would be great to repeat this study at a site which provides enough ‘classic’ archaeological evidence, so the blind test of the phytolith and geochemical data could be backed up and the method could be validated. This logic should be at least mentioned in the discussion so that readers understand the limitation of the approach.

> The main aim of the discussed method is to enable the incorporation of different types of evidence to aid archaeological interpretation, the verification of geoarchaeological methods is not really the focus of this paper. The statistical model presented is in itself a method of validation, using the analysis results to verify a prior hypothesis (which is based on archaeological evidence - the interpretation of features in the field). The (un)certainty of the reclassification is briefly addressed in lines 499-502 (revised version). 

Applying the model to a site that has more substantial evidence would not necessarily test the statistical model, but the evidence used in it. Nevertheless, there are many possible interesting follow up studies, this is now referred to in the discussion (lines 513-516). And it would definitely be beneficial for geoarchaeological studies to explore ethnographic and more substantial settings. But perhaps testing this would require at least some uncertainty regarding the interpretation of the site in the first place. If it is too straightforward, the evidence might simply confirm the prior probability. And if it does not – what does that mean?

---

## [Decision Letter · Decision Letter 1]

24 Feb 2021

A model based on Bayesian confirmation and machine learning algorithms to aid archaeological interpretation by integrating incompatible data

PONE-D-20-17716R1

Dear Dr. Vos,

We’re pleased to inform you that your manuscript has been judged scientifically suitable for publication and will be formally accepted for publication once it meets all outstanding technical requirements.

Kind regards,

Peter F. Biehl, PhD

Academic Editor

PLOS ONE

Additional Editor Comments (optional):

Reviewers' comments:

Reviewer's Responses to Questions

**Comments to the Author**

1. If the authors have adequately addressed your comments raised in a previous round of review and you feel that this manuscript is now acceptable for publication, you may indicate that here to bypass the “Comments to the Author” section, enter your conflict of interest statement in the “Confidential to Editor” section, and submit your "Accept" recommendation.

Reviewer #3: All comments have been addressed

2. Is the manuscript technically sound, and do the data support the conclusions?

Reviewer #3: Yes

3. Has the statistical analysis been performed appropriately and rigorously? 

Reviewer #3: Yes

4. Have the authors made all data underlying the findings in their manuscript fully available?

Reviewer #3: Yes

5. Is the manuscript presented in an intelligible fashion and written in standard English?

Reviewer #3: Yes

6. Review Comments to the Author

Reviewer #3: This is a well-written and focused paper that is very easy to follow. The authors present their topic and case study in a natural and straightforward way, and are very conscientious about using unassuming language and explicitly introducing and defining concepts. As the authors suggest, this is important for addressing a general audience; however, I would go a step further and say it's a good practice in addressing ANY audience.

So far as I can tell, the statistical methods are sound. The assumptions in the model are fairly simplistic, but it doesn't have to be complex. If anything, I appreciate that it provides a clear illustration of the methods used without the burden of a highly complex set of assumptions or a huge mass of data. The data used are all provided and steps are clearly outlined as to how to replicate the analysis.

The concerns of previous reviewers seem to have been adequately addressed.

7. PLOS authors have the option to publish the peer review history of their article (what does this mean?). If published, this will include your full peer review and any attached files.

Reviewer #3: No

---

## [Editor Report · Acceptance letter]

2 Mar 2021

PONE-D-20-17716R1 

A model based on Bayesian confirmation and machine learning algorithms to aid archaeological interpretation by integrating incompatible data 

Dear Dr. Vos:

I'm pleased to inform you that your manuscript has been deemed suitable for publication in PLOS ONE. Congratulations! Your manuscript is now with our production department. 

Kind regards, 

on behalf of

Dr. Peter F. Biehl 

Academic Editor

PLOS ONE